# Development and application of a tsunami fragility curve of the 2015 tsunami in Coquimbo, Chile

Rafael Aránguiz[1,2], Luisa Urra[3], Ryo Okuwaki[4], Yuji Yagi[5]

[1]Department of Civil Engineering, Universidad Católica de la Santísima Concepción
5  [2]National Research Center for Integrated Natural Disaster Management CONICYT/FONDAP/1511007 (CIGIDEN)
[3]Laboratory of Remote Sensing and Geoinformatics for Disaster Management, International Research Institute of Disaster Science, Tohoku University, Japan
[4]Graduate School of Life and Environmental Sciences, University of Tsukuba, Japan
[5]Faculty of Life and Environmental Sciences, University of Tsukuba, Japan

10  *Correspondence to*: Rafael Aránguiz (raranguiz@ucsc.cl)

**Abstract.** The last earthquake that affected the city of Coquimbo took place in September 2015 and had a magnitude of Mw 8.3, resulting in localized damage in low-lying areas of the city. In addition, another seismic gap north of the 2015 earthquake rupture area has been identified; therefore, a significant earthquake (Mw 8.2 to 8.5) and tsunami could occur in the near future. The present paper develops the tsunami fragility curve for the city of Coquimbo based on field survey data 15 and tsunami numerical simulations. The inundation depth of the 2015 Chile tsunami in Coquimbo was estimated by means of numerical simulation with the NEOWAVE model and 5 nested grids with a maximum grid resolution of 10 m. The fragility curve exhibited behavior similar to that of other curves in flat areas in Japan, where little damage was observed at relatively high inundation depths. In addition, it was observed that Coquimbo experienced less damage than Dichato (Chile); in fact, at an inundation depth of 2 m, Dichato had a ~75% probability of damage, while Coquimbo proved to have only a 20  20% probability. The new fragility curve was used to estimate the damage by possible future tsunamis in the area. The damage assessment showed that ~50% of the structures in the low-lying area of Coquimbo have a high probability of damage in case of a tsunami generated off the coast of the study area if the city is rebuilt with the same types of structures.

## 1 Introduction

On 16 September 2015 a Mw 8.3 earthquake took place off the coast of the Coquimbo Region (USGS: 25  http://earthquake.usgs.gov/earthquakes/eventpage/us20003k7a#executive). The earthquake generated a tsunami that inundated low-lying areas of the city of Coquimbo, with runup reaching up to 6.4 m and a penetration distance of up to 700 m (Aránguiz et al., 2016; Contreras-López et al., 2016), resulting in reports of significant damage to houses and public infrastructure (Contreras-López et al., 2016). This earthquake filled the seismic gap that had existed since at least the last significant earthquake along the Coquimbo-Illapel seismic region in 1943 (Melgar et al., 2016; Ye et al. 2016). However, the 30  region just north of the 2015 rupture area has not experienced significant seismic activity since the 1922 Mw 8.3 event (Melgar et al., 2016; Ye et al., 2016). Thus, it is recommended that reconstruction plans and new tsunami mitigation

measures consider potential impacts due to possible future tsunamis generated north of the 2015 Illapel earthquake rupture zone.

With regard to the assessment of structural damage within the exposed area against potential tsunami hazard, two different approaches were identified. Damage can be estimated deterministically based on the forces acting on a single structure (Nandasena et al. 2012; Nistor et al. 2009; Shimozono & Sato, 2016; Wei et al., 2015); however, such an analysis could be extremely time-consuming and impractical for an entire city due to the high-resolution numerical simulations (~2 m) that are required. Alternatively, the assessment of structural damage could be done probabilistically by means of fragility curves (Koshimura et al. 2009; Koshimura et al., 2009(1); Suppasri et al, 2011). Tsunami fragility curves represent the probability of damage to structures in relation to a tsunami intensity measure, such as the inundation depth, current velocity or hydrodynamic force (Koshimura et al. 2009), although a fully probabilistic approach may use a wide range of possible scenarios; thus, both hazard assessment and damage assessment are probabilistic (Park et al., 2017b). A classical approach uses linear models with ordinary least-square methods and aggregated data. This methodology has been applied to obtain empirical tsunami fragility curves for Banda Aceh in Indonesia (Koshimura et al. 2009) and Thailand (Suppasri et al., 2011) after the 2004 Indian Ocean Tsunami. The same methodology was applied to areas affected by the 2009 Samoa tsunami (Gokon et al., 2014). In a similar manner, this method was applied in Japan after the 2011 Great East Japan tsunami, allowing several fragility curves that considered several damage levels and different building materials to be obtained (Suppasri et al., 2013). After the 2010 Chile tsunami, Mas et al. (2012) developed the first tsunami fragility curve in Chile for masonry and mixed structures in Dichato. In recent years, new methodologies have been proposed for the development of tsunami fragility curves that use disaggregated data and different classes of models such as the Generalized Linear Model, Generalized Additive Model and Non-Parametric Model (Charvet et al., 2017; Charvet et al., 2015; Macabuag et al., 2016). These new methodologies propose a more comprehensive analysis in order to select appropriate statistical models and identify which tsunami intensity measure gives the best representation of the observed damage data (Macabuag et al., 2016). Even though the use of different classes of models could offer an improvement over the ordinary least-square method, there is no quantifiable assessment of the effect of data aggregation and linear model assumption violation on the predictive power of a model (Macabuag et al., 2016). For example, the fragility curves developed by Suppasri et al. (2013) have been applied to building damage estimation in Napier, New Zealand (Fraser et al., 2014) and both building damage and economic loss estimation in Seaside, Oregon (Wiebe & Cox, 2014). The former study also applied the fragility curves of Dichato, Chile (Mas et al., 2012), and American Samoa (Gokon et al., 2014).

Tsunami fragility curves are obtained for a given area under a given scenario; therefore, they may not be applicable to other areas of interest since the tsunami characteristics and building materials may differ (Koshimura et al. 2009; Suppasri et al., 2011). For example, buildings along the Sanriku ria coast in Japan experienced greater damage than structures located in the plains of Sendai (Suppasri et al., 2013; Suppasri et al., 2012); thus, De Risi et al. (2017) analyzed the influence of tsunami velocity on structural damage on ria-type and plain-type coasts. They found that while flow velocity improves the fragility models, the two coastal typologies should be considered separately when velocity is included in the analysis. Moreover,

Song et al. (2017) used a bivariate intensity measure to evaluate tsunami losses, such that both flow velocity and inundation depth are analyzed. They found that flow velocity is important for buildings located less than 1 km from the coastline. In addition, they found that reinforced concrete buildings are the most sensitive to the incorporation of velocity, while wood structures exhibit no sensitivity to this variable.

The Coquimbo area provides a good opportunity to develop a fragility curve and assess potential tsunami impact since the tsunami in 2015 did not damage all structures and some of the damaged structures have been repaired or rebuilt on their original sites. This study develops an empirical fragility curve for the Coquimbo area using field survey data and numerical simulation of the 2015 Chile tsunami. In addition, we estimated the probability of structural damage for a deterministic tsunami scenario using the Coquimbo fragility curve. Section 2 gives a description of the study area, with a short review of

the local seismicity. Section 3 presents the methodology of the fragility curve development, which includes a comparison with existing tsunami fragility curves. Section 4 presents an application of the fragility curves. Finally, section 5 gives the main conclusions of the present research.

## 2 Study area

The city of Coquimbo is located on the southern shore of Coquimbo Bay (29.96°S). The Coquimbo area was mentioned by

the conquistadors as a good place for a port and the location became important due to the natural protection it offered against south-west swell waves, which provided good conditions for its use as a port starting in the XIX century. Coquimbo Bay is open to the northwest and characterized by a lowland topography with a long, flat, sandy beach (Aránguiz et al., 2016), similar to the coastal plains of Sendai.  Like all coastal cities in Chile, Coquimbo is located over the subduction zone of the Nazca Plate beneath the South American plate (18°-44°S). The convergence rate of the plates is 68 mm/year along the Chile

subduction zone and large seismic events take place every 10 years on average (Métois et al. 2016). In fact, 3 events over magnitude 8.0 have taken place in the last 6 years, namely, the 2010 Maule (34-38°S) , 2014 Iquique (19-20°S) and 2015 Illapel (30-32°S) earthquakes.

Figure 1 shows the seismic events recorded in the Coquimbo area. The oldest record of a tsunami is that of the 1730 event. This earthquake generated a destructive tsunami that destroyed Valparaiso and Concepción and flooded low-lying areas in

Japan (Cisternas et al., 2011). The tsunami destroyed several ranches on the shore of Coquimbo (Soloviev & Go, 1975). Although the 1880 and 1943 earthquakes are considered to be similar in size (Nishenko, 1985), it is observed that the behaviors of the tsunamis generated by these events seem to be different. While the former generated large columns of water that resulted in the anchor chain of a ship snapping in Coquimbo (Soloviev & Go, 1975) and a deep submarine cable breaking off the coast near the mouth of the Limarí River (Lomnitz, 2004), the latter generated a minor tsunami that

damaged fishing boats in Los Vilos and raised the water level by 80 cm in Valparaiso (Soloviev & Go, 1975), while no tsunami was reported in Coquimbo.  Conversely, the 2015 tsunami reached up to 4.75 m at the Coquimbo tide gauge, with a runup of 6.4 m (Aránguiz et al., 2016; Contreras-López et al., 2016). Moreover, a maximum tsunami amplitude of 2 m was

observed at the Valparaiso tide gauge (Aránguiz et al., 2016). The main reason behind this is that the 1943 event broke the deepest portion of the subduction interface, while the 2015 Illapel earthquake had a shallower rupture area and a larger

magnitude (Fuentes et al., 2016; Okuwaki et al., 2016), resulting in a larger initial tsunami amplitude (Aránguiz et al., 2016). The largest tsunami ever recorded in Coquimbo took place in 1922. It arrived in Coquimbo two hours after the earthquake, with three large waves observed, the third of which was the largest, with a maximum inundation height of 7 m and an inland penetration of 2 km. The part of the city located on the southern shore of Coquimbo Bay was totally destroyed by both the water and tsunami debris (Soloviev & Go, 1975).  In a similar manner, the tsunami reached inundation heights of up to 9 m

at Chañaral and 6-7 m at Caldera. The tsunami was also observed in Japan, with maximum amplitudes ranging from 60 to 70 cm (Carvajal et al., 2017; Soloviev & Go, 1975), which is similar to the amplitudes of the 2015 event (80 cm), but larger than those of the 1943 event, which were 10-25 cm (Beck et al., 1998). Another significant event was the 1849 earthquake, which generated a localized tsunami that mainly affected Coquimbo. The tsunami arrived 10 to 30 min after the earthquake, penetrated 300 m horizontally and rose 5 m above the high tide mark (Soloviev & Go, 1975).

## 3 Development of the fragility curve

The development of the fragility functions in the present work required three main steps: first, data collection regarding building damage levels in the Coquimbo area and tsunami inundation heights for numerical modeling validation; second, selection of a rupture model of the 2015 Illapel earthquake and validation of the tsunami inundation heights for estimation of tsunami inundation depth; and third, GIS analysis and statistical analysis for correlation between damage level and simulated

tsunami inundation depth.

### 3.1 Building damage and tsunami inundation data

Only 5 to 7 days after the 2015 event, a team surveyed the affected area and collected more than 40 inundation height, inundation depth and tsunami runup measurements in the Coquimbo inundation area. The field measurements followed established post-tsunami survey procedures (Dengler et al., 2003; Dominey-Howes et al., 2012; Synolakis & Okal, 2005)

and were corrected for tide level at the time of maximum inundation. At the same time, 585 structures within the inundation area were identified and classified as mixed structures made of wood and masonry (568), reinforced concrete buildings of eight or more stories (4) and very light structures that did not meet minimal building standards (13). The present analysis considered the mixed structures only; therefore, the reinforced concrete and light structures were removed from the fragility curve analysis. Typical structures within the inundated area of Coquimbo have one story and are made of masonry, though

there are some two-story buildings made of both masonry (the first floor) and wood (the second floor). In order to facilitate the comparison with existing fragility curves (e.g., Dichato) all data were combined in a single category: mixed structures. Figure 2 shows typical mixed structures and inundation depth marks surveyed in Coquimbo immediately after the 2015 tsunami.  Figures 2a and b show masonry houses that were not damaged by the tsunami despite inundation depths that

ranged from 1.5 to 2 m. Meanwhile, Figures 2c and d show houses with moderate to major damage, possibly to be used after major repairs. In fact, the house in Figure 2c was being repaired at the time of the field survey and the gray wall in the corner had been built a few days earlier. Meanwhile, the house in Figure 2d was abandoned, since all interior walls, windows, doors and the roof were destroyed and major repairs and retrofitting will be needed. Figure 2e shows a destroyed structure with its interior walls and roof completely removed, while Figure 2f shows the remaining foundation of a washed-away structure. Even though the damage to the structure could be due to both the earthquake and tsunami, it was observed that damage due to the earthquake was limited (Candia et al., 2017) and the structures most affected by the earthquake were made of adobe (Fernández et al., 2017). In addition, the authors had the opportunity to compare damage to inundated and non-inundated houses in Coquimbo in order to verify that the structural damage to inundated houses was due to the tsunami. In order to avoid categorizing light damage (due to the earthquake) as tsunami damage, a two-level damage scale was used. Thus, the present work assumed that the damage to flooded structures was due only to the tsunami.

In addition, the two-level damage scale was used due to the small amount of inundated structures (568) and for comparison with the existing fragility curve of Dichato (Mas et al., 2012), which has only two damage levels. The first level, called "not-destroyed," included structures with no damage or minor to major damage, corresponding to levels 1, 2 and 3 given by Suppasri et al. (2013). These damage levels indicate that there is slight to severe damage to non-structural components; therefore, it would be possible to use the structures after moderate to major repairs (Figures 2a, b and c). The other damage level, called "destroyed," included damage levels 4 to 6 according to Suppasri et al. (2013), i.e., structures that underwent severe damage to walls or columns or that had completely collapsed (Figures 2 d, e and f).

Previous works carried out damage inspections using satellite images and field surveys (Koshimura et al., 2009; Mas et al., 2012; Suppasri et al., 2011); however, the satellite image method assumes that buildings with intact roofs are "not destroyed" (Suppasri et al., 2011), and severe damage to columns or interior walls may not be observed (Mas et al., 2012), as in the case of the houses shown in Figures 2c and d. Therefore, the present work employed damage detection based on field surveys only. Figure 3a shows the surveyed buildings and the damage levels assigned to the 568 mixed structures. The four reinforced concrete buildings (R.C.) and the 13 light structures (L.S.) that did not meet minimal building standards are also included in the figure. Figure 3b shows the inundation height and runup measurements recorded during the field survey. It is observed that the maximum inundation height was reached in the corner, where the coastal road and the railway converge. Most of the damaged structures were identified in that location as well.

### 3.2 Tsunami inundation depth

Tsunami inundation depth was estimated as the difference between tsunami inundation height and ground elevation. Since the inundation heights were measured at few locations across the inundation area and there is a lack of tsunami traces in the wetland, interpolation of tsunami height may not be suitable; therefore, the tsunami heights were obtained from tsunami numerical simulation of the 2015 event. We tested four available finite-fault models, namely, Li et al. (2016), Ruiz et al. (2016), Okuwaki et al. (2016) and Shrivastava et al. (2016), and the best fit was selected according to tide gauges in

Coquimbo and Valparaiso and DART buoy 32402. Once the best slip model was selected, we used the field measurements of inundation height and runup to select an appropriate dry-land roughness coefficient. The model proposed by Li et al. (2016) is obtained from iterative modeling of teleseismic body waves as well as tsunami records at DART buoys. Since the magnitude of the proposed model is Mw 8.21, the slip distribution was multiplied by a factor of 1.38; thus, all events have the same magnitude: 8.3.

The tsunami initial condition was estimated to be equal to the seafloor displacement. In addition, the vertical displacement from each subfault was computed using a kinematic solution of the planar fault model of Okada (Okada, 1985). The numerical simulations were carried out with the Non-hydrostatic Evolution of Ocean WAVEs model (NEOWAVE) (Yamazaki et al. 2011; Yamazaki et al., 2009). This model is a staggered finite-difference model that solves the nonlinear shallow water equation and uses a vertical velocity term to account for weakly dispersive waves. The model generates the tsunami initial condition, propagation and inundation by means of several nested grids of different resolutions. The present research used 5 nested grids, as shown in Figure 4. The level-1 grid describes tsunami propagation from generation to the continental shelf and to the Pacific Ocean at a resolution of 2 arcmin (~3600m). This grid was generated from 30 arcmin GEBCO data. The level-2 and level-3 grids were built from nautical charts 4100, 4112 and 4113, and had a resolution of 30 and 6 arcsec, respectively. The level-4 grid covered Coquimbo Bay and was built from nautical chart 4111, and had a resolution of 1 arcsec (~30m). Finally, the level-5 grid had a resolution of 1/3 arcsec (~10m) and was built from bathymetry from nautical chart 4111 and topography from a DTM with a 2-m resolution provided by the Coquimbo office of the Ministry of Public Works (MINVU). The topography used high-resolution data; thus, the most important features, such as the coastal road embankment, railway, river and wetland, are well represented (see Figure 4, grid 5). Numerical simulations in Valparaiso involved four nested grids with a maximum grid resolution of 1 arcsec (~30 m).

The roughness coefficient was defined as n=0.025 on the seabed, as recommended for tsunamis (Bricker et al., 2015; Kotani et al., 1998); however, we tested several roughness coefficient values in coastal, wetland and urban areas in order to obtain the best fit of tsunami inundation height. The validation of the numerical simulation was performed using the Root Mean Square Error and the parameters $K$ and $\kappa$ given by equations (1) and (2) (Aida, 1978) . The variable $K_i$ is defined as $K_i = x_i / y_i$, where $x_i$ and $y_i$ are recorded and computed tsunami heights, respectively. The Japan Society of Civil Engineering provides guidelines, which recommend that $0.95 < K < 1.05$ and $\kappa < 1.45$ for there to be "good agreement" (Aida, 1978; Gokon et al., 2014).

$$\log K = \frac{1}{n} \sum_{i=1}^{n} \log K_i, \tag{1}$$

$$\log \kappa = \sqrt{\frac{1}{n} \sum_{i=1}^{n} \left( \log K_i \right)^2 - \left( \log K \right)^2}, \tag{2}$$

Figure 5 shows the tsunami initial conditions of the four slip models and the tsunami waveforms over an elapsed time of 4 hours at three selected gauges, namely, Coquimbo, Valparaiso and DART buoy 32402. Even though the modified Li et al. (2016) model overestimates the maximum amplitude at the DART buoy, the simulation exhibits a very good agreement with the tsunami record in Coquimbo. When the Mw 8.3 models proposed by Ruiz et al. (2016) and Shrivastava et al. (2016) were analyzed, it was possible to observe a good agreement at the DART buoy and Valparaiso tide gauge, although the amplitude in Coquimbo is underestimated by more than a meter. The Okuwaki et al. (2016) model overestimates both the DART buoy and Valparaiso tide gauge, despite the second tsunami wave reaching a similar amplitude in Coquimbo. Nevertheless, the maximum tsunami amplitude is underestimated. Therefore, the modified Li et al. (2016) model was selected to assess the suitable Manning roughness coefficient.

Figure 6 shows the inundation area and tsunami inundation height results obtained from the numerical simulations of the Li et al. (2016) model, with four different roughness coefficients. The tested coefficients are n=0.025 for coastal and riverine areas, 0.04 and 0.05 for low-density urban areas and 0.06 for medium-density urban areas (Bricker et al., 2015; Kotani et al., 1998). From the figure, it is possible to observe that the best fit is obtained for n=0.025, which resulted in $K=1.05$ and $\kappa<1.45$, corresponding to "good agreement." For higher roughness coefficients, the tsunami inundation heights are underestimated. In addition, the larger the coefficient, the smaller the inundation area. This behavior could be explained by the fact that a significant part of the flooded area is a wetland and the developed area is rather small, with a low-density residential distribution. Thus, the inundation depth is computed from the inundation area given by the modified Li et al. (2016) slip model, with a roughness coefficient of n=0.025.

**3.3 Fragility curve**

The construction of a fragility curve requires a correlation between the structural damage level and a tsunami intensity measure, such as the inundation depth, current velocity or hydrodynamic force. To this end, we used the classical approach with aggregated data and a least-square fit (Koshimura et al., 2009), in which a sample size is defined such that each range of the tsunami intensity measure includes the defined number of structures. Then the damage probability is calculated by counting the number of destroyed or not-destroyed structures for each range of the intensity measure. Finally, the fragility function is developed through regression analysis of the discrete set of damage probabilities and the tsunami intensity measure. Therefore, it is assumed that the cumulative probability P of damage follows the standardized normal or lognormal distribution function given in equation (3). $\Phi$ is the distribution function, $x$ is the hydrodynamic feature of the tsunami and $\mu$ and $\sigma$ are the mean and standard deviation of $x$, respectively. The values of $\mu$ and $\sigma$ are calculated by means of least-squares fitting of $x$ and the inverse of $\Phi$, $(\Phi^{-1})$ on normal paper given by equation (4).

$$P(x)=\Phi\left[\frac{x-\mu}{\sigma}\right], \tag{3}$$

$$x = \sigma \, \Phi^{-1} + \mu,$$ (4)

The hydrodynamic force per unit width (kN/m) acting on a structure is computed as the drag force given by equation (5), where the drag coefficient is assumed to be $C_D = 1.0$ for simplicity, $\rho$ is the density of sea water (1,025 $kg/m^3$), $U$ is the flow velocity (m/s) and $h$ is the inundation depth (m).

$$F = \frac{1}{2} C_D \rho h U^2$$ (5)

Figure 7 shows the results of the simulated tsunami intensity measures. It can be observed that the topography plays an important role in tsunami inundation, as the maximum inundation depth values (Figure 7 a) occur at the beach and wetland,

while developed areas behind the railway and areas distant from the shore present low inundation depths. In a similar manner, high velocities occur close to the sites of rapid topographic changes (Figure 7 b), such as the lee side of the coastal road, while low velocities are observed within the developed area under analysis (<3 m/s). Since hydrodynamic force is a combination of both inundation depth and flow velocity (Figure 7 c), the developed area behind the railway presents low force as well. Figure 8 shows the results of the tsunami fragility curves of Coquimbo for inundation depth, flow velocity and

hydrodynamic force. The sample size was defined to be 40 structures; thus, 15 ranges were used. Figure 8a shows the histogram, while Figure 8c shows the relationship between damage probability and inundation depth (upper panel), flow velocity (central panel) and hydrodynamic force (lower panel), with the solid line representing the best-fit curve of the plot. The fragility curves were estimated by means of regression analysis, as shown in Figure 8b. The statistical parameters of the developed fragility functions are shown in Table 1. In Figure 8 it is possible to observe that inundation depths lower than 1.5

240 m did not generate damage to the surveyed structures and the damage probability of the curve is less than 10%. Moreover, the fragility curve shows that inundation depths higher than 4 m could result in a 100% probability of severe damage to mixed structures in Coquimbo. With regard to the flow velocity, it is observed that most of the simulated data are in the range of 0 to 2.5 m/s, with a damage probability of less than 40%. In a similar manner, a hydrodynamic force lower than 2.5 kN/m proves to result in a damage probability of less than 20%.

Since the 2015 tsunami had a moderate impact, with low inundation depths and flow velocities in developed areas, it becomes very important to assess the tsunami damage due to possible events taking place in the same rupture area as that of the 1922 earthquake, since large inundation depths were reported there (see section 2).

## 3.4 Comparison with existing fragility curves

This section compares the fragility curve obtained in Coquimbo with curves obtained in other places after recent events. The statistical parameters of existing fragility curves are shown in Table 2. One curve is that of Okushiri, Japan, which was obtained for wooden structures after the 1993 tsunami event. The analysis included 523 houses and a range of approximately 50 structures (Suppasri et al., 2012). In a similar manner, the fragility curve of Dichato, Chile, involved 915 mixed-material structures and a range of 50 structures after the 2010 Chile tsunami (Mas et al., 2012). A more comprehensive analysis was conducted in Banda Aceh, Indonesia, after the Indian Ocean Tsunami (Koshimura et al., 2009). This case involved 48,910 structures made of wood, timber and lightly reinforced concrete constructions, with a range of 1,000 structures. The proposed curves were constructed for inundation depth, flow velocity and hydrodynamic force. After the 2009 Samoa event, Gokon et al. (2014) developed a fragility curve for mixed structures, which included wood, masonry and reinforced concrete, for the same three tsunami-intensity measures as in the previously mentioned study. Similarly, the fragility curves of Thailand were developed for two provinces, namely, Phang Nga and Phuket, with 2,508 and 1,033 structures, respectively. In addition, all data were combined in order to develop a fragility curve for mixed-material structures and inundation depth (Suppasri et al., 2011). Figure 9a shows a comparison of the Coquimbo fragility curve with 2-level damage curves of Dichato, Okushiri, Banda Aceh, American Samoa and Thailand. It is seen that Coquimbo experienced less damage than Dichato and Okushiri at inundation depths lower than 3 m. In fact, at an inundation depth of 2 m, Dichato and Okushiri have a 68-75% probability of damage, while in Coquimbo the probability is only 20%. The high probability of damage in Dichato and Okushiri could be due to the large number of structures made of wood and lightweight materials with little ability to withstand tsunami flows (Mas et al., 2012). Even though the building materials in Coquimbo are similar, it is observed in Figure 7 b that distance from the shore and the railway embankment decrease flow velocity and thus tsunami energy; therefore, the same inundation depth generates less damage to structures. In a similar manner, the fragility curve for mixed-material structures in Thailand shows a high probability of damage at an inundation depth of 2 m (~50%), but a 100% probability of damage is reached at inundation depths higher than 8 m. In the case of Banda Aceh, the curve shows a low probability of damage (<20%) at an inundation depth of 2 m, which is comparable to Coquimbo; however, the damage probability in Coquimbo increases rapidly as the inundation depth increases, reaching 100% at an inundation depth of only 4 m, which could be a result of most of the houses having only 1 or 2 stories (see Figure 2). In addition, it was observed in Banda Aceh that structures were quite vulnerable when flow velocity exceeded 2.5 m/s, with a damage probability of 60%, and a 100% probability of damage at velocities larger than 4 m/s (Koshimura et al., 2009). These results are in good agreement with the Coquimbo fragility curve. Moreover, the topography of Banda Aceh is characterized by low land with an elevation of around 3 m, which is also similar to Coquimbo. With regard to American Samoa, the curve shows a low probability of damage at inundation depths lower than 2 m; it begins to increase to up to 80% when the inundation depth reaches 6 m. It is important to mention that the Samoa fragility curves were developed considering different types of structures, including wood, brick and reinforced concrete. In addition, the fragility curve as a function of flow velocity shows

significant damage (~50%) at velocities of 2 m/s, and only an 80% probability of damage at velocities as high as 8 m/s (Gokon et al., 2014). Since all types of structures are analyzed in a single curve, it is believed that low velocities would easily cause damage to wooden structures, while damage to reinforced concrete structures would require higher inundation depths and flow velocities. The relatively high damage probability at low inundation depths could be also due to the ria-type coast of American Samoa (Gokon et al., 2014).

Figures 9b and c show the comparison of the Coquimbo fragility curve with the curves given by Suppasri et al. (2013) for wooden and mixed-material structures in Japan, respectively. The study considered more than 250,000 damaged buildings surveyed after the 2011 Great East Japan Tsunami and made it possible to analyze different damage levels and building materials. In general, it is seen that wooden and mixed structures in Japan have similar behavior. If damage level 4 (complete destruction) is analyzed, the damage probability is higher than in Coquimbo at an inundation depth lower than 2 m. Wooden and mixed structures in Japan present a relatively high probability of complete destruction (level 4), ranging from 50 to 60%, while in Coquimbo it is only 20%.

Another group of fragility curves for wooden and mixed structures – shown in Figures 9d and e, respectively – were obtained from survey data of the 2011 Japan tsunami in the Sendai and Ishinomaki plains (Suppasri et al., 2012). The curves show that structures located in flat areas were less impacted by the tsunami despite significant inundation depths, in contrast to what happened in areas with ria topography, such as the Sanriku coast (Suppasri et al., 2013; Suppasri et al., 2012) and semi-closed bays such as Dichato (Mas et al., 2012). This behavior is in good agreement with damage observed in the Coquimbo area, where the flat nature of the area and distance from the shore could decrease tsunami impact. Thus, based on the influence of inundation depth and flow velocity on tsunami damage, De Risi et al. (2017) proposed the development of vulnerability models related to specific topographic contexts, such as plain-type or ria-type coasts. They found that ria-type coasts experience greater damage probability than plain-type coasts at the same inundation depth.

It is noteworthy that the Coquimbo fragility curve for destruction or complete damage overlaps with the minor-damage-level curve for wood and mixed-material houses in flat areas in Japan (Figure 9d and e). A possible explanation is that houses in Japan are relatively new and built according to strict construction standards (Suppasri et al., 2012), in contrast to what was observed in Coquimbo, where old houses are found (See Figure 2), although it could also be due to the local topographic features of Coquimbo. This finding suggests that both topography and structure quality should be considered in tsunami damage estimation.

## 4 Application of fragility curve to tsunami damage estimation

This section presents an example of the use of fragility curves to estimate tsunami damage through a deterministic tsunami scenario in Coquimbo. We first define a tsunami scenario, then we run the numerical simulation to obtain the inundation depth and, finally, we estimate the tsunami damage in Coquimbo. Since earthquake damage in the Coquimbo Region was

limited in 2015 (Candia et al., 2017b; Fernández et al., 2017), it is assumed that the damage to structures is due exclusively
to the tsunami.

### 4.1. Tsunami source model

Based on Figure 1, three possible segments can be defined, namely, the Copiapó-Coquimbo, Coquimbo-Illapel and Illapel-Constitución regions. However, events in the Illapel-Constitución  region, including those of 1822 and 1906, have never
generated a tsunami in Coquimbo (Soloviev & Go, 1975), and only the 1730 event, which ruptured the Coquimbo-Illapel segment, generated a tsunami in the area of interest (Cisternas et al., 2011); therefore, possible tsunamis generated in the Valparaiso segment were not considered in the present analysis. In a similar manner, earthquakes on the Coquimbo-Illapel segment were not considered because the 2015 Illapel earthquake filled the seismic gap that had existed since the last major earthquake in 1943 or earlier events (Ye et al., 2016); thus, no significant earthquakes that generate significant tsunamis
could take place there in the near future. Conversely, the northern segment has presented no relevant seismic activity since 1922, i.e., 95 years before 2017 (see Figure 1); moreover, the previous significant event took place in 1819 (73 years before the 1922 event). Therefore, the Copiapó-Coquimbo segment is of particular interest regarding possible future earthquakes and tsunamis in Coquimbo.

It is important to note that the small event in 1849 (magnitude 7.5, according to Lomnitz (2004)) generated a 5-m tsunami in
Coquimbo. Despite the small earthquake magnitude and large tsunami runup of the event, there is no scientific evidence that a tsunami-earthquake occurred. In addition, the 1922 Atacama event had a complex source of three time-clustered shocks (Beck et al., 1998). Therefore, it seemed reasonable to separate the northern segments into two different seismic regions, with one segment covering Copiapó to Punta Choros (Figure 10b) and the second segment from Punta Choros to Ovalle (Figure 10a), which also coincides with the estimated rupture length of the 1849 event (see Figure 1).
Either a probabilistic or deterministic approach could be used for the tsunami hazard assessment and damage estimation. While the former takes into account many uncertainties related to generation, propagation and inundation  (Cheung et al., 2011; Geist & Parsons, 2006; Heidarzadeh & Kijko, 2011; Horspool et al., 2014; Park & Cox, 2016), the latter uses credible worst-case scenarios based on historical events (Aránguiz et al., 2014; Mitsoudis et al., 2012; Wijetunge, 2012). However, the coupling coefficient could be used to assess the shape of possible future deterministic earthquakes (Métois et al., 2016;
Pulido et al., 2015), since reasonable heterogeneous slip models could be predicted by the degree of interseismic locking (Calisto et al., 2016; Gonzalez-Carrasco et al., 2015). Thus, the slip distribution $S$ at arbitrary space $\xi$ is represented as given by equation (6):

$$S(\xi)=\int_{t_0}^{t_1} C(\xi,t)V(\xi)dt - \sum_j \left(s_j(\xi)+p_j(\xi)\right), \tag{6}$$

where $C$ is the interseismic coupling, ranging from 0 to 1. The interseismic coupling  model adopted in this study is from Métois et al., (2016), which is derived from inverting Global Positioning System (GPS) measurements along the Chilean margin (18–38°S) that have been made by international teams since the early 1990s (see Métois et al. (2016) and references therein). It provides a reasonable estimate of the degree of locking between the Nazca and the South American plates, indicating strong coupling along the scenario source regions (see Figures 10d to f). $V$ is the plate convergence rate at $\xi$, derived from the NNR-Nuvel1A model (DeMets et al., 1994), and $t_0$ and $t_1$ delimit the interseismic period for integration. $s_j$ is the slip of the small event ($4.8 \leq \mathrm{Mw} \leq 7.9$) at the $j$th location, which is listed in the Global Centroid Moment Tensor (GCMT) Catalog (http://www.globalcmt.org/CMTsearch.html; see Figure 10e), and $p_j$ is the post-seismic slip following $s_j$. Each amount of slip $s_j$ is calculated based on the seismic moment obtained by the GCMT and the empirical relationship between rupture area and the moment magnitude introduced by Wells & Coppersmith (1994). The rigidity modulus for the calculation of moment magnitude of each $s_j$ is computed with the layered, near-source structure adopted in the source study by Okuwaki et al. (2016). We eliminated the Mw 8.3 2015 Illapel earthquake from the GCMT list and instead considered its contribution to the scenario source models with the inverted slip model developed by Okuwaki et al. (2016) in equation (6) (Fig. 10). The slip motion of $S$ is assumed to be pure thrust against the subducting plate motion. Note that $C$ is constant against time and the post-seismic slip $p_j$ is not considered in the present analysis; thus, it is possible that the scenario source models will slightly overestimate $S$.

The variable slip distribution was obtained from the heterogeneous interseismic coupling $C$. Time intervals for the integral of equation (6) are assumed to be 94 years (1922 to 2016). Each segment is subdivided into 10 km x 10 km sub-space knots for 150 x 160 km2 and 180 x 160 km2 source areas for S1 and S2, respectively. While the magnitude of the event related to segment S1 is Mw 8.2, the magnitude of the S2 event is Mw 8.4. If both segments are considered together (S3=S1+S2), the total magnitude is Mw 8.5. The strike and dip angles for the scenario source geometry are assumed to be constant based on the subducting slab geometry of the Slab 1.0 model (Hayes et al., 2012): (Strike, Dip) = (2.7°, 15.0°) for S1 and (Strike, Dip) = (16.0°, 15.0°) for S2. The fault geometry and characteristic source parameters, as well as complete model parameters for each scenario source model, are available from the authors upon request.

## 4.2 Numerical simulation of proposed tsunami scenario

The computation covered an elapsed time of 6 hours with output intervals of 1 min. Figure 11 shows the main results and the three different tsunami scenario combinations. The upper row shows the results for segment S1 (Mw 8.2) and the middle row shows the results for segment S2 (Mw 8.4), while the lower row shows the results for the combined scenario of S1 and S2 (Mw 8.5). In addition, the left column shows the vertical displacement of the seafloor, the middle column shows the

maximum inundation depth and the right column shows the tsunami wave form at the Coquimbo tide gauge over an elapsed
time of 4 hours (240 min). It is observed that segment S2 (Mw 8.4) generated lower inundation depths than segment S1 (Mw 8.2), which can be explained by the fact that the strike angle and the coastal morphology cause the tsunami to be propagated toward the north and not directly toward Coquimbo Bay. Meanwhile, the tsunami generated by segment S1, the second wave of which is the largest, propagates directly toward Coquimbo Bay. It is possible to observe that the maximum inundation depths reached up to 5 m in developed areas and along the coastline. Moreover, it is interesting that the Mw 8.5 event, as a
combination of S1 and S2 (lower row in Figure 11), generated lower inundation depths than segment S1 alone. This can be explained by the fact that the maximum tsunami amplitude of each individual event does not occur at the same time; thus, the segment S2 tsunami decreases the maximum amplitude of the segment S1 tsunami. Larger tsunami amplitudes could result from a time gap between the segment S1 and S2 events such that the maximum tsunami waves coincide. Nevertheless, this analysis is beyond the scope of the present paper.

### 4.3 Damage to structures

The previous section demonstrated that the combination of S1 and S2 rupturing at the same time generated lower inundation heights than the S1 event alone; therefore, the damage to structures is assessed for segment S1 only, i.e., a tsunami generated by a Mw 8.2 earthquake off the coast of Coquimbo that generates inundation heights lower than 5 m. Figure 12 shows the
results for each tsunami intensity measure, namely, inundation depth, flow velocity and hydrodynamic force (upper row). In addition, the lower row in Figure 12 shows the difference between the maximum tsunami intensity measures given by the S1 scenario and those of the simulated 2015 tsunami event (Figure 7). This figure allows areas with a greater increase in tsunami intensity measure and, therefore, higher damage probability, to be identified.

In order to determine a high or low probability of damage to a given structure, first, latitude and longitude coordinates are assigned to each structure within the inundation area, and the maximum inundation depths given by the tsunami numerical simulation at the location of each structure are exported to GIS. Second, the inundation depth database is divided into several ranges, with 40 samples in each range, and the mean value of each range is intersected with the fragility curve given in Figure 8c in order to define the damage probability for each range. For simplicity, and similar to previous studies (Fraser et
al., 2014; Wiebe & Cox, 2014), we used only the fragility curve generated as a function of the inundation depth. Third, the damage probability given in the previous step is assumed to be equal to the percentage of structures with a high probability of damage within each range. To make this determination, the inundation depths for each range are arranged in descending order and the structures outside of that percentage (with the lowest inundation depth within the range) are assumed to have a low probability of damage.

Figure 13a shows the low-lying area of the city of Coquimbo and the computed inundation depth given by the numerical simulation of scenario S1. 646 mixed-material structures were identified within the inundation area, and they are colored according to inundation depth level. Figure 13b shows the result of the damage estimation. It was found that 321 structures, i.e., 49.6% of the flooded structures, have a high probability of damage, a figure that is much higher than the 20% surveyed right after the 2015 tsunami. As expected, the structures behind the railway embankment and wetland would experience less damage than those located close to the shore.

Due to the high probability of damage to houses located near the shore, it is recommended that any reconstruction plan or future tsunami mitigation measures consider the fact that high tsunami inundation depths (5-8 m) could be generated in this area. After the 2011 Japan tsunami, it has been demonstrated that comprehensive urban planning is the key point for avoiding future disasters, such that the best approach to decrease tsunami risk is an integration of structural/non-structural means of coastal protection and land-use management as a strategy with multiple lines of defense (Strusińska-Correia, 2017). In addition, the most important lessons from the 2011 Japan tsunami include methods to strengthen coastal defense structures, evacuation buildings and coastal forests (Suppasri et al., 2016). Thus, Coquimbo seems to be an interesting case study, since the coastal road, wetland and railway partly fulfill the structural requirements of a multilayer tsunami countermeasure, and it would be necessary to implement more comprehensive non-structural countermeasures in the future. In a local context, Khew et al. (2015) found that the tsunami countermeasures implemented in the Greater Concepcion area after the 2010 Chile tsunami, such as hard infrastructure, contributed positively to the recovery of economic and social resilience, although it was found that new elevated housing decreased social resilience. Moreover, it is recommended that governmental and business structures be effectively decentralized such that local conditions are successfully incorporated into the design of hard infrastructure for tsunami mitigation (Khew et al., 2015). Finally, it was also found that tsunami mitigation measures implemented in Dichato after the 2010 Chile tsunami did not decrease tsunami risk, as some vulnerability variables (housing conditions, low household incomes and limited knowledge of tsunami events) are still at the same level (Martínez et al., 2017). Therefore, non-structural mitigation measures should play an important role in effectively decreasing tsunami risk in the future.

## 5 Conclusions

Numerical simulations of the 2015 Chile tsunami proved to be in good agreement with field survey data in Coquimbo. A Coquimbo fragility curve was developed with two-level classification of structural damage, namely, "not-destroyed" and "destroyed." The Coquimbo fragility curve shows a low probability of damage, 20%, at a relatively high inundation depth (2m), in contrast to what was observed in another Chilean town, Dichato, where a 68% probability of damage resulted from the same inundation depth. This result is in good agreement with fragility curves in the Sendai and Ishinomaki plains in Japan, in that tsunami energy decreased and less damage was observed.

The fragility curve may be used to estimate possible future tsunami damage in the Coquimbo area and other places with similar topography and building materials. In Coquimbo, it was found that a Magnitude Mw 8.2 earthquake off the coast of the city could generate a destructive tsunami with inundation depths of up to 5 m. The assessment of tsunami damage with the fragility curve demonstrated that ~50% of the assessed structures have a high probability of damage if the reconstruction is carried out with the same types of structures, which is greater than the damage caused by the 2015 tsunami (20%). Therefore, tsunami mitigation measures and the reconstruction plan should consider potential tsunami damage due to a future earthquake off the coast of Coquimbo. It is recommended that new land-use policies be implemented in order to regulate the types of structures being built in the inundation area. In addition, based on previous experience in Japan and Chile, new tsunami mitigation measures must consider a combination of both structural and non-structural tsunami countermeasures in order to effectively decrease tsunami risk in Coquimbo in the future.

**Acknowledgements**

The authors would like to thank CONICYT (Chile) for its FONDAP 15110017 and FONDECYT 11140424 grants, as well as the Research and Innovation Department (Dirección de Investigación e Innovación) of the Universidad Católica Ssma. Concepción. Special thanks to those who contributed to the collection of field data: Associate Professor Dr. Enrique Muñoz and students Evelyn Pedrero, Evans Aravena and Diego Espinoza. Thanks to the Ministry of Housing for providing us with topography data. Finally, thanks to the two anonymous reviewers, who significantly helped us improve the manuscript.

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

**Table 1. Statistical parameters for developed fragility curves obtained from a normal distribution.**

| Tsunami intensity measure | $\mu$ | $\sigma$ | $R^2$ |
|---|---|---|---|
| Inundation depth (m) | 2.4395 | 0.5537 | 0.8524 |
| Flow velocity (m/s) | 2.5268 | 0.6421 | 0.8580 |
| Hydrodynamic force (kN/m) | 4.2564 | 1.7055 | 0.7512 |


**Table 2. Summary of statistical parameters and damage levels for empirical fragility curves (Mas et al., 2012; Suppasri et al., 2013; Suppasri et al., 2012) including the current case of Coquimbo. $\mu$ and $\sigma$ are statistical parameters for normal distribution, while $\mu'$ and $\sigma'$ are the same parameters for lognormal distribution**

| Event | Location | Structure type | Damage level | No. structures inspected | $\mu$ | $\sigma$ | $\mu'$ | $\sigma'$ | $R^2$ |
|---|---|---|---|---|---|---|---|---|---|
| Chile (2010) | Dichato-Chile | Wood, masonry, mixed | Not destroyed/ Destroyed | 915 | | | 0.092 | 1.272 | 0.86 |
| Japan (2011) | Okushiri-Japan | Wood | Not destroyed/ Destroyed | 523 | | | 0.216 | 0.736 | 0.82 |
| Indian Ocean (2004) | Banda Aceh-Indonecia | Wood, RC | Not destroyed/ Destroyed | 48,910 | 2.985 | 1.117 | | | 0.99 |
| Indian Ocean (2004) | Thailand | Mixed | Not destroyed/Destroyed | 3541 | | | 0.747 | 0.984 | 0.88 |
| Samoa (2009) | American Samoa | Wood, brick and RC. | Not destroyed/Destroyed | | | | 1.17 | 0.69 | 0.89 |
| Japan (2011) | Hokkaido, Aomori, Iwate, Miyagi, Fukushima, Ibaraki, Chiba | Wood | Level 1 | 251,000 (total) | | | -2.1216 | 1.2261 | 0.98 |
| | | | Level 2 | | | | -0.9338 | 0.9144 | 0.98 |
| | | | Level 3 | | | | -0.040 | 0.7276 | 0.98 |
| | | | Level 4 | | | | 0.6721 | 0.4985 | 0.98 |
| | | | Level 5 | | | | 0.7825 | 0.5559 | 0.98 |
| | | | Level 6 | | | | 1.2094 | 0.5247 | 0.97 |
| Japan (2011) | Hokkaido, Aomori, Iwate, Miyagi, Fukushima, Ibaraki, Chiba | Mixed | Level 1 | 251,000 (total) | | | -2.4562 | 1.4874 | 0.99 |
| | | | Level 2 | | | | -1.1373 | 1.115 | 0.96 |
| | | | Level 3 | | | | -0.0756 | 0.8277 | 0.97 |
| | | | Level 4 | | | | 0.5316 | 0.6235 | 0.91 |
| | | | Level 5 | | | | 0.8336 | 0.6077 | 0.97 |
| | | | Level 6 | | | | 1.2244 | 0.5723 | 0.98 |
| Japan (2011) | Ishinomaki and Sendai plains | Wood | Minor | 150 | 2.4409 | 0.6409 | | | 0.95 |
| | | | Moderate | | 2.9028 | 0.6777 | | | 0.94 |
| | | | Major | | 3.8458 | 0.8516 | | | 0.95 |
| | | | Complete | | 4.2243 | 1.0159 | | | 0.80 |
| Japan (2011) | Ishinomaki and Sendai plains | Mixed | Minor | 189 | 2.4954 | 0.8249 | | | 0.81 |
| | | | Moderate | | 3.2550 | 1.0647 | | | 0.80 |
| | | | Major | | 4.4355 | 1.3068 | | | 0.83 |
| | | | Complete | | 5.0620 | 1.4872 | | | 0.84 |


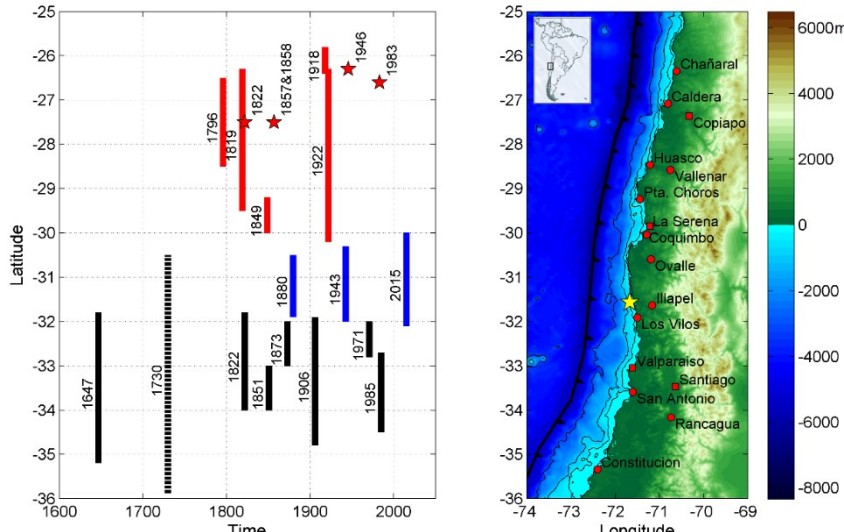

**Figure 1: Seismicity of central Chile. Left: Space-time plot of large earthquakes along central Chile. Red bars are the events along the Copiapó-Coquimbo region and the red stars represent smaller seismic events. The blue bars are events along the Coquimbo-Illapel seismic region, while the black lines represent events along the Los Vilos-Constitución segment. The dashed line is the large event of 1730, which ruptured both the Los Vilos-Constitución and Coquimbo-Illapel segments. (Beck et al., 1998; Lomnitz, 2004; Métois et al., 2016; Nishenko, 1985). Right: Map showing the cities and towns mentioned in the text. The yellow star represents the epicenter of the 2015 Illapel Earthquake. The thin black lines are isobaths at water depths of 200, 1000 and 3000 m. The thick black line is the Peru-Chile trench.**

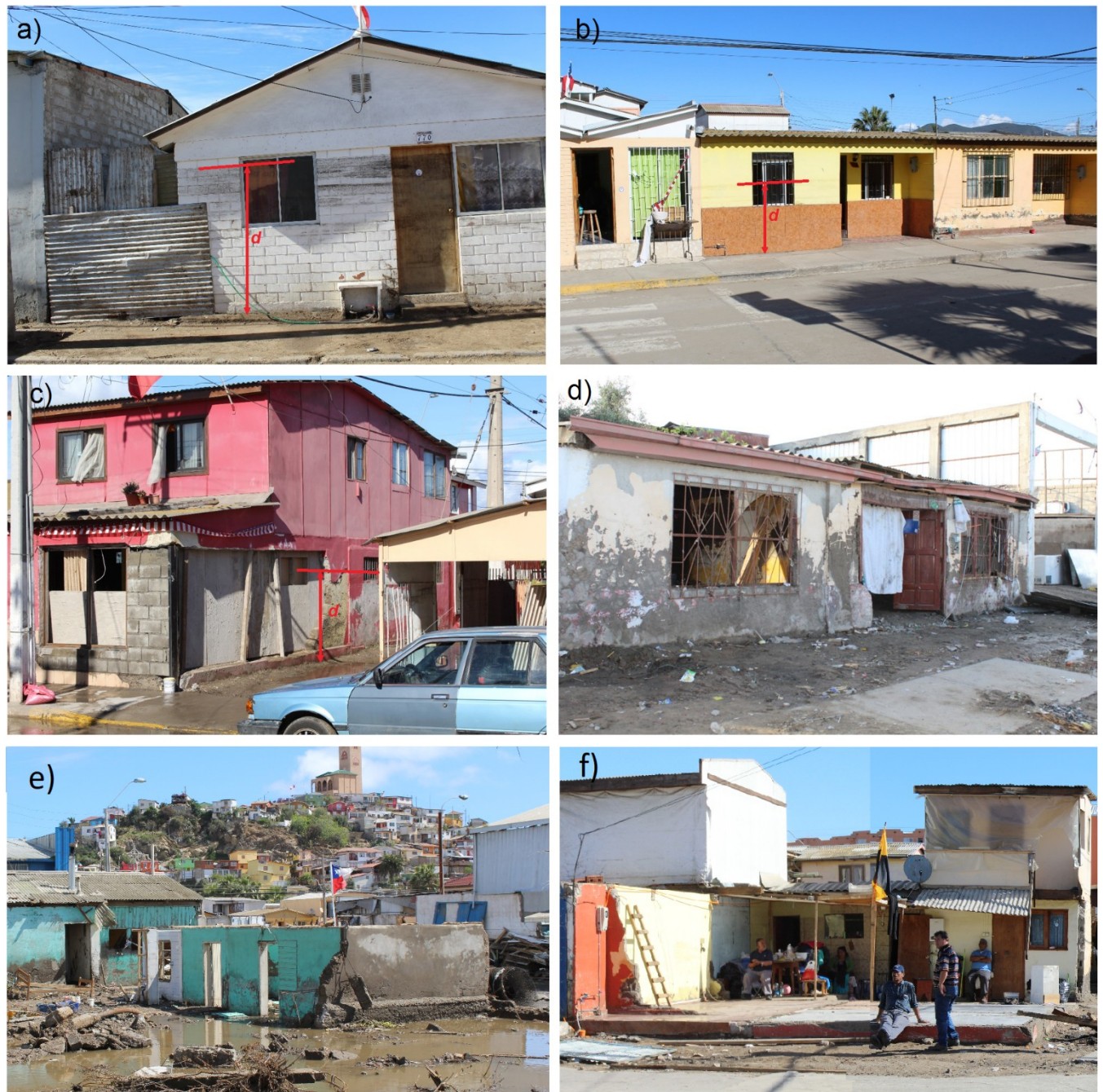


**Figure 2: Photographs of structures undamaged and masonry houses damaged by the September 16<sup>th</sup> 2015 tsunami in the Coquimbo area. The red letter d indicates the observed tsunami inundation depth. All photos were taken on September 22<sup>nd</sup> of 2015.**

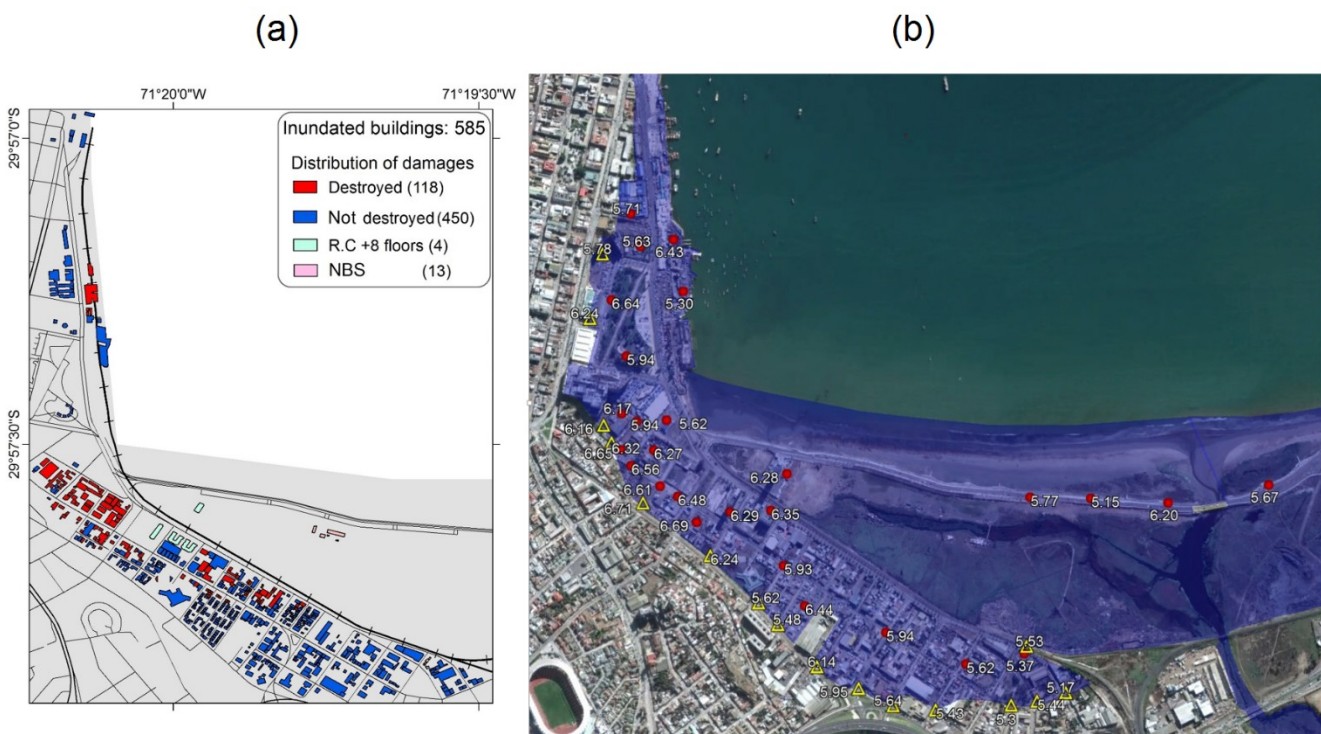

**Figure 3. a) Surveyed damage to structures due to the 2015 tsunami; RC: reinforced concrete structures, L.S.: Light structures. b) Coquimbo inundated area (Aránguiz et al., 2016) and survey data. Red circles represent inundation measures and yellow triangles tsunami runup.**

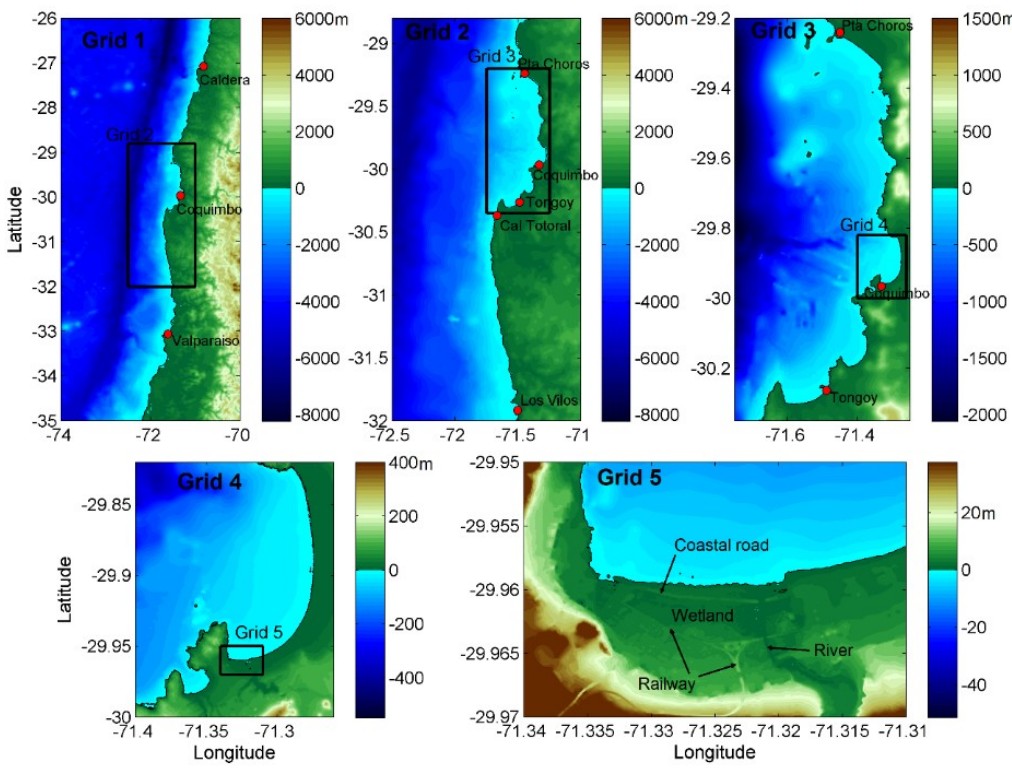

Figure 4. Model setting and nested computational grids for Coquimbo.

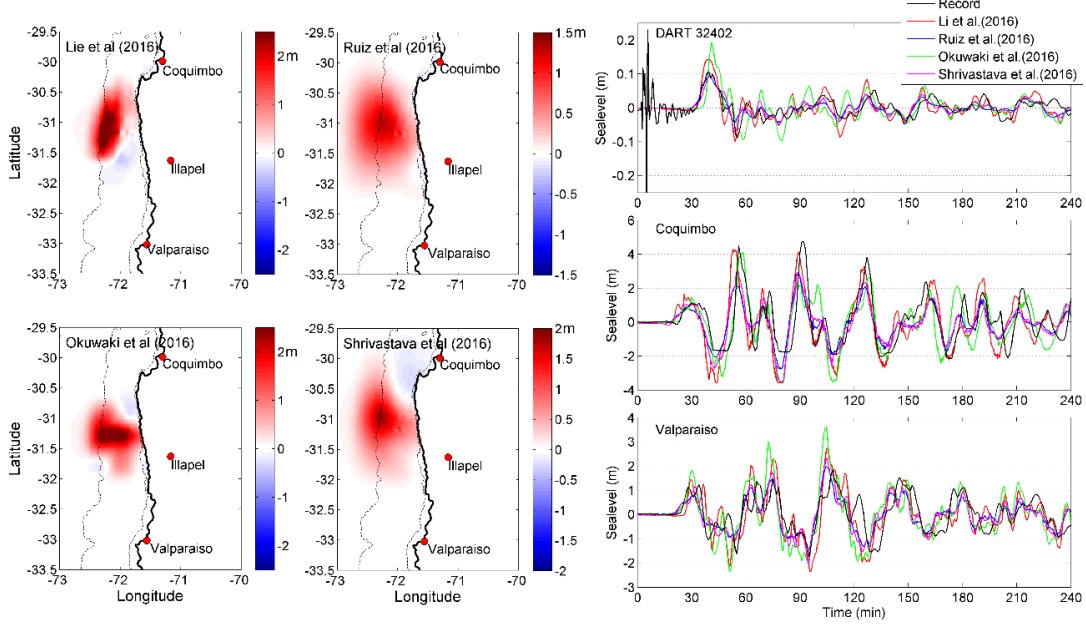

**Figure 5. Tsunami initial conditions of four source models and comparison of tsunami records with simulated tsunami waveforms at DART 32402, Coquimbo and Valparaiso.**

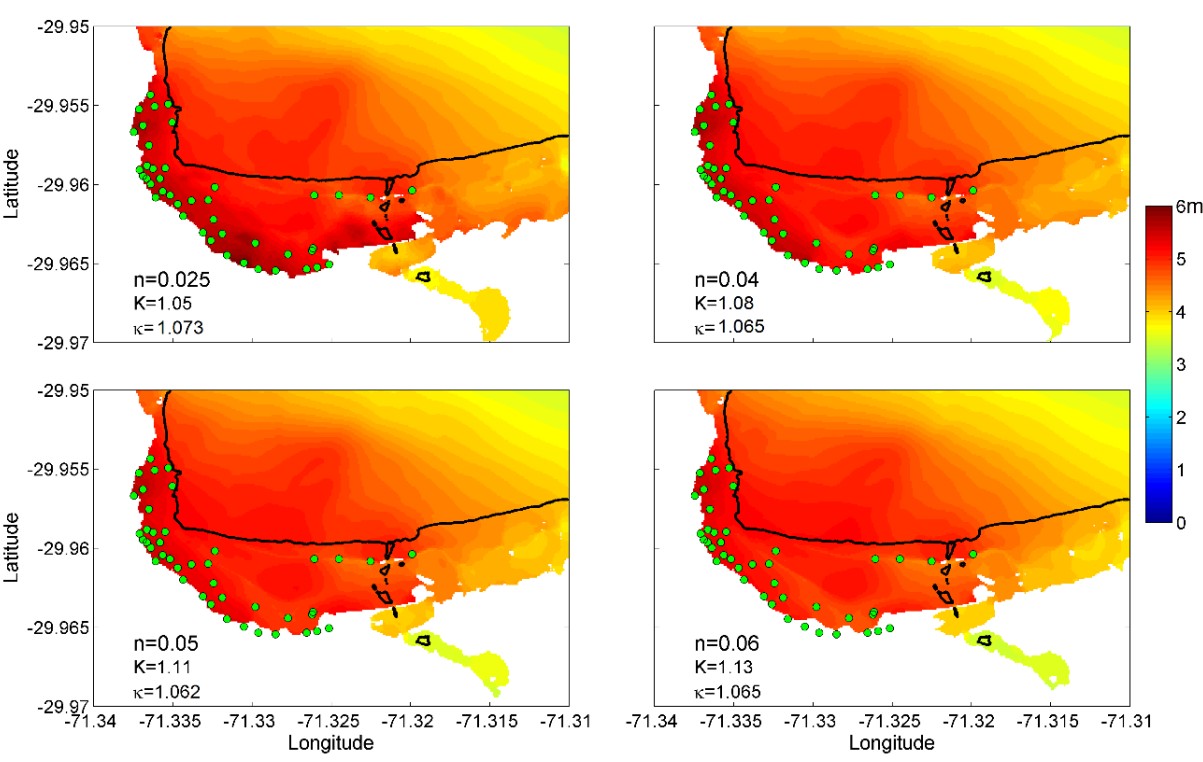

**Figure 6. Tsunami inundation heights obtained with the modified Li et al. (2016) source model and four different Manning coefficients, n=0.025, 0.04, 0.05, 0.06. The parameters of root mean square error, $K$ and $\kappa$ area also shown.**

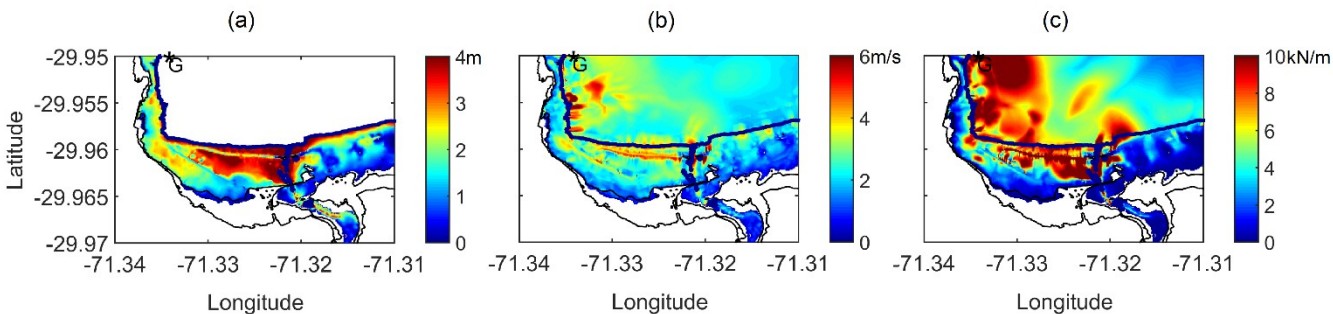

**Figure 7. Results of tsunami numerical simulations for each intensity measure. a) Inundation depth, b) flow velocity**
**and c) Hydrodynamic force.**

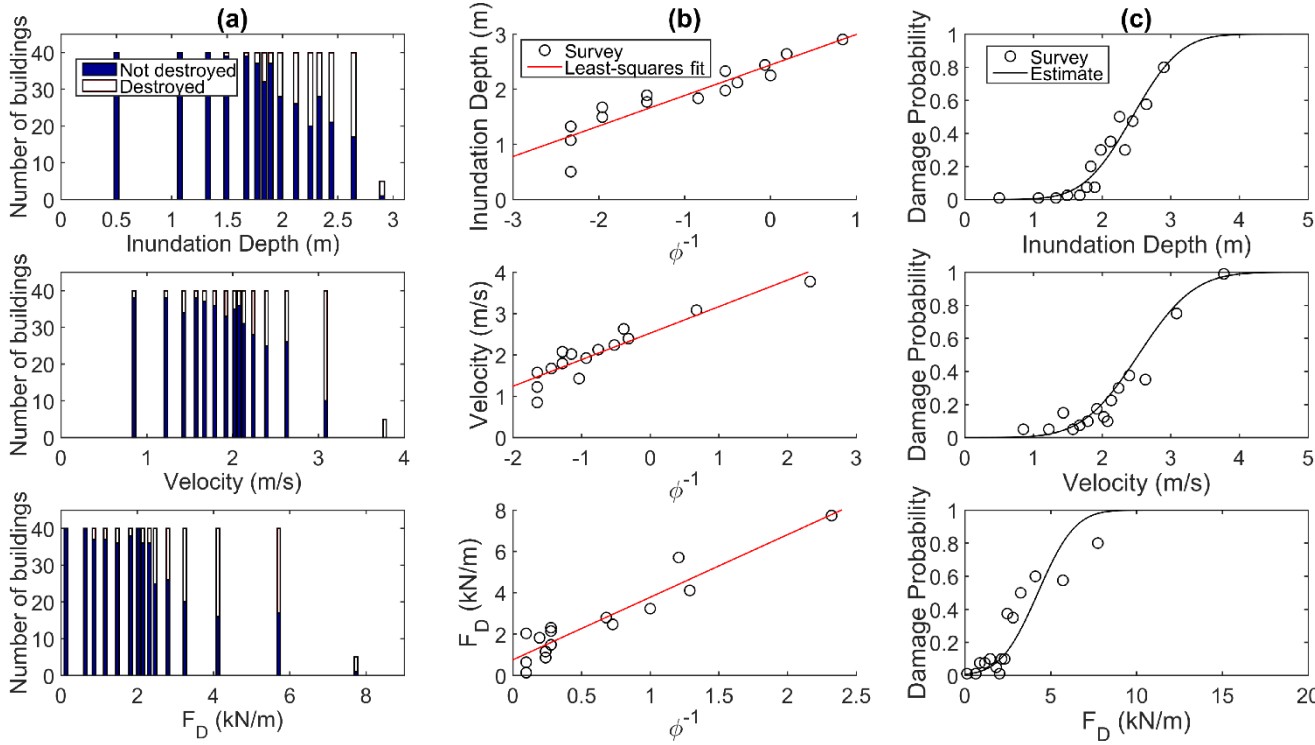

**Figure 8. Developing the tsunami fragility curve. a) Histogram of the number of destroyed and not-destroyed structures in terms of the tsunami intensity measures within the inundation area. b) Data plotted on normal probability paper and least-squares fit. c) Fragility function for building damage in terms of the tsunami intensity measures; the solid line is the best-fit curve of the plot (o: the distribution of damage probability).**

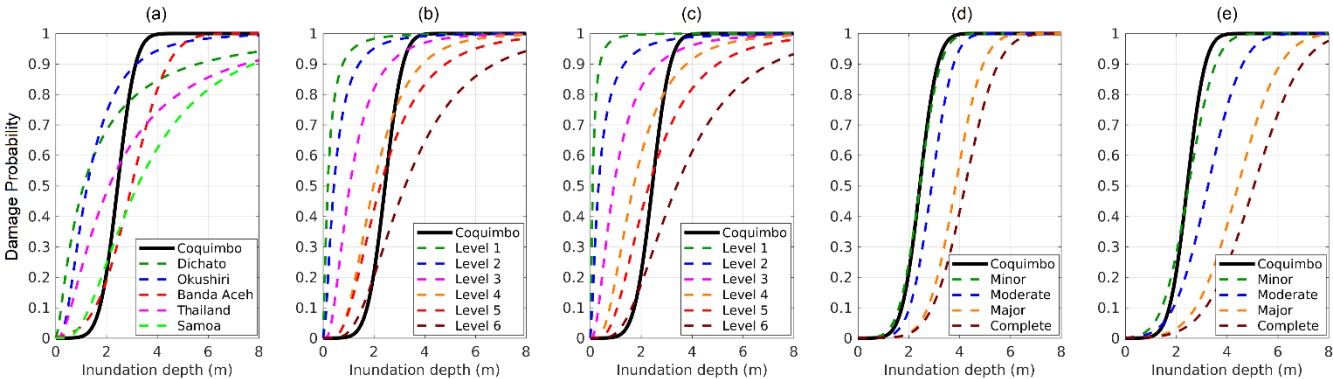

**Figure 9. Tsunami fragility curves for damage probability developed for other locations and different damage levels. a) Two levels of damage obtained for three different cities in Chile, Japan and Indonesia. b)  Six damage levels for wooden structures given by Suppasri et al. (2013). c) Six damage levels for mixed-material structures by Suppasri et al. (2013). d) Four damage levels for wooden houses given by Suppasri et al. (2012). e) Four damage levels for mixed-material structures given by Suppasri et al. (2012).**

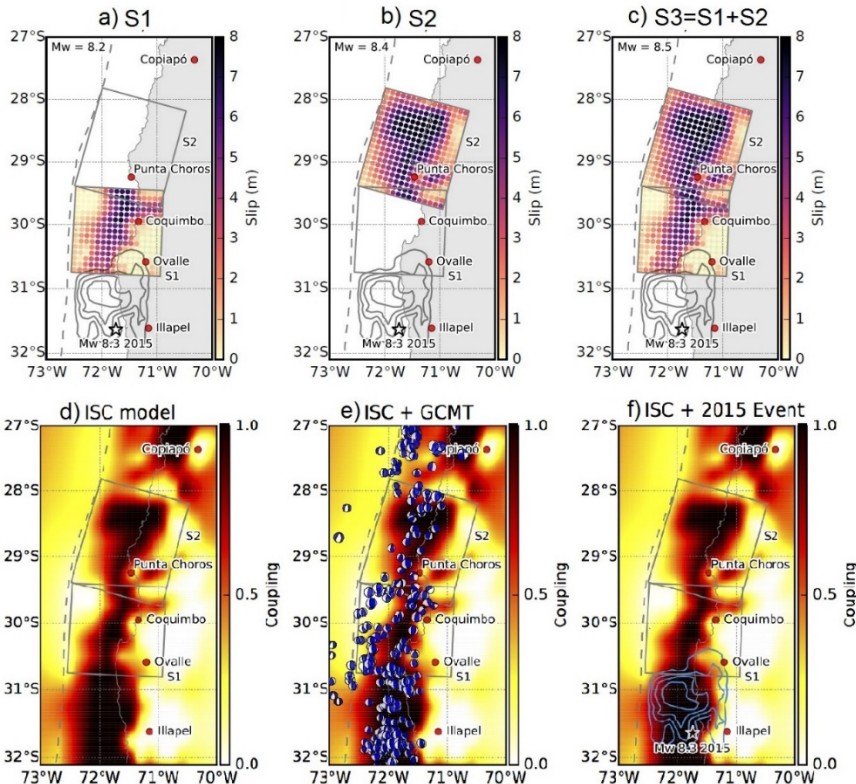


**Figure 10. Upper panel: Slip distributions along scenario source models. The grey rectangles outline each scenario source segment. The moment magnitude for each scenario source model is denoted in the top left of the corresponding panel. Lower panel: (left) the ISC model from Métois et al. (2016), (center) GCMT solutions and (right) the inverted slip model from Okuwaki et al. (2016), which were used to construct the scenario source models. The star denotes the epicenter of the 2015 Illapel earthquake determined**
**by the National Seismological Center (CSN, for its initials in Spanish). The blue contours delimit the inverted slip distribution every 2.08 m for the 2015 Illapel earthquake (Okuwaki et al., 2016).**

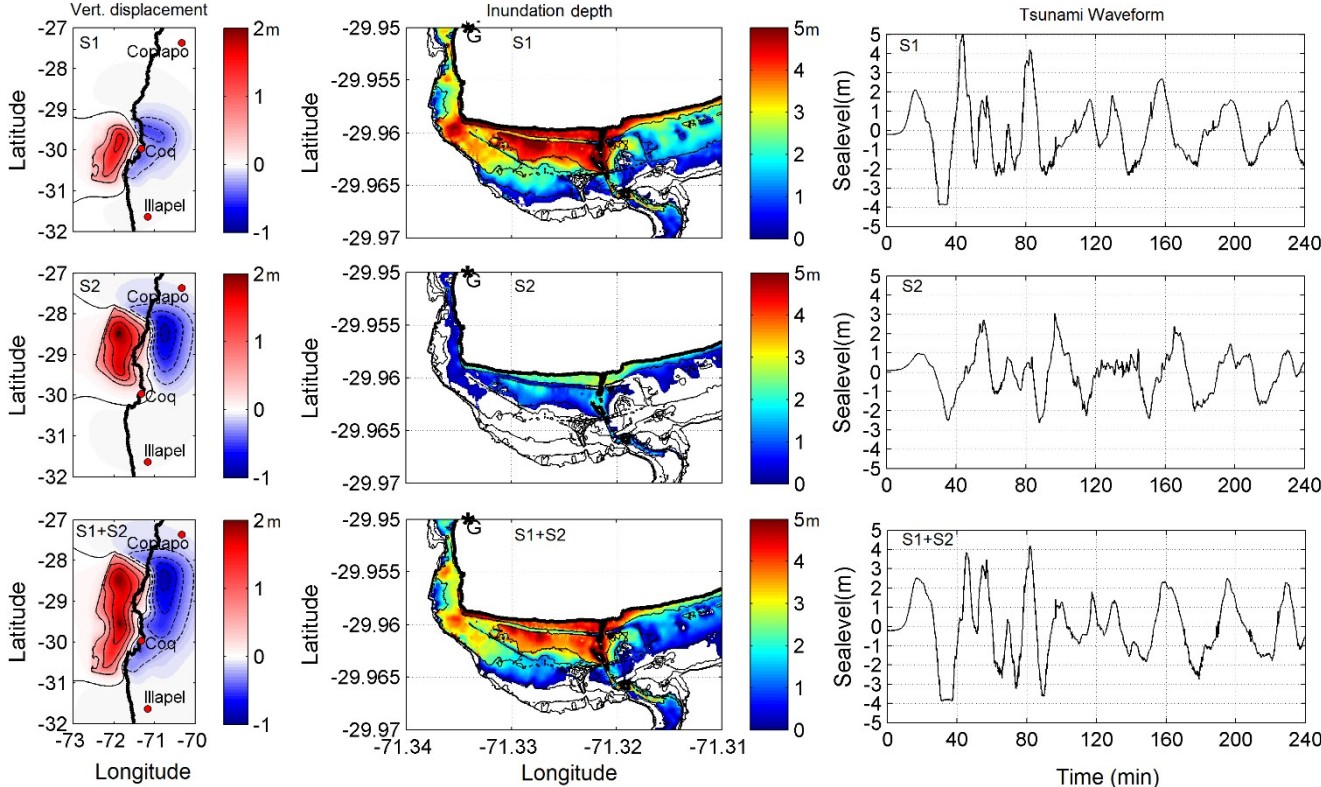

**Figure 11. Results of tsunami numerical simulations for case 1 and the three scenarios, S1, S2 and S1+S2. Left column: vertical seafloor displacement. Central column: maximum inundation depth; the asterisk indicates the location of the tide gauge and the thin black lines represent the contour lines every 2 m. Right column: tsunami wave form over an elapsed time of 4 h at the Coquimbo tide gauge G.**

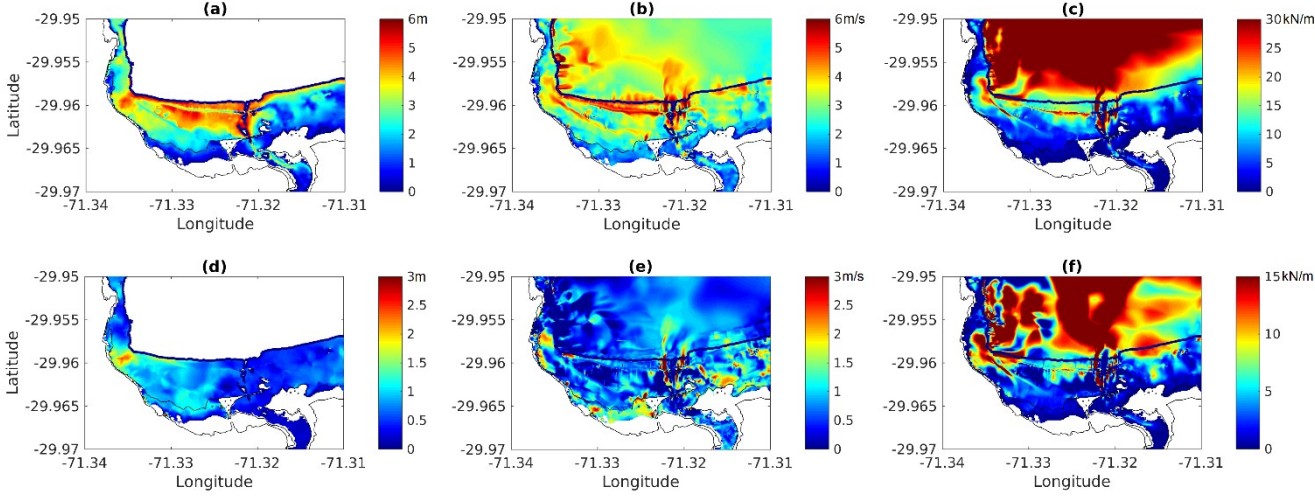


**Figure 12. Results of tsunami numerical simulation of S1 event (Mw 8.2). a) Inundation depth b) Flow velocity, c) Hydrodynamic force, d) Increase in inundation height compared to 2015 Coquimbo tsunami. e) Increase in Flow velocity compared to 2015 Coquimbo tsunami. f) Increase in Hydrodynamic force compared to 2015 Coquimbo tsunami.**

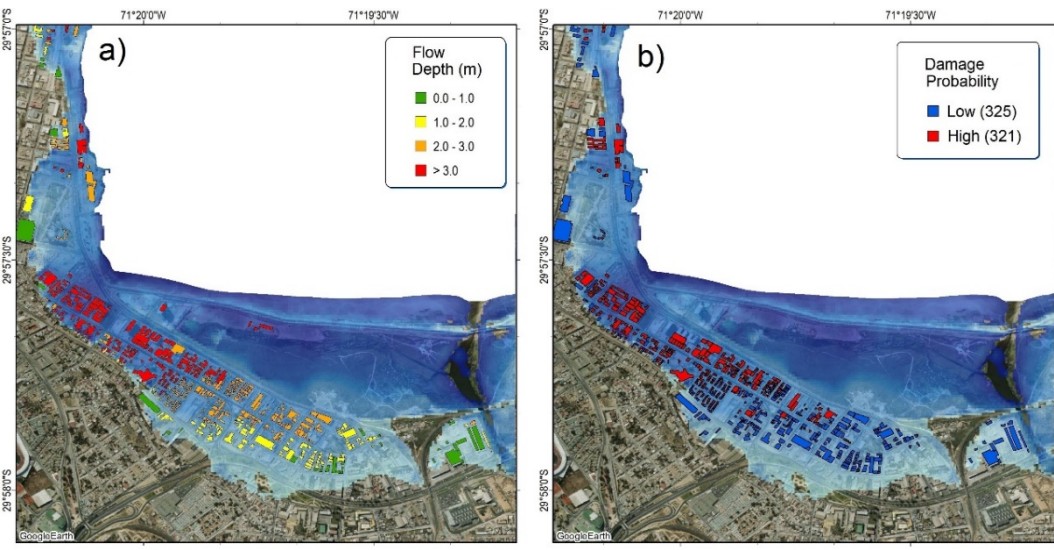


**Figure 13. a) Tsunami inundation map and inundation depth on structures. b) Tsunami inundation map and low and high probabilities of damage to the flooded structures.**