# Peer review of "Development and application of a tsunami fragility curve of the 2015 tsunami in Coquimbo, Chile"

_Natural Hazards and Earth System Sciences, 2017_

## Referee Comment (RC1) · Anonymous Referee #1 · 22 Nov 2017

I congratulated the effort of the authors as it seems that the manuscript was revised before submitting to NHESS. However, there are still some main augments (see sections 3.1-3.3) . Please see all of my comments below I believe that they will help improving the manuscript.

Title: Please consider adding the part "application of the developed fragility curves" in your title. Abstract: Please also specify the earthquake magnitude of both the 2015 and future events as well as add a short explanation about the comparison of the developed curves against Dichato. 1 Introduction: Please make sure that you use the same name, 2015 Illapel earthquake (same in Fig.1) throughout the paper. In the last

sentence of this section, your objective is only about two sentences. I would suggest to write them clearly using bullets and separate them from the literature review. There are some recently published works about fragility curves and their application that should be mentioned, Song et al. (2017), Charvet et al. (2017), Macabuag et al. (2016), Suppasri et al. (2016), Fraser et al. (2014) and Wiebe and Cox (2014) 2 Study area: Please also clearly state the topographical characteristic of the study area, plain area similar to Sendai? 3 Developing fragility curve: What kind of the field data used as I can only see the recorded waveforms? Did you also measure inundation heights/depths from the field survey? If so why didn't you used them for your model verification in addition to the recorded waveforms? 3.1 Data: What is the average numbers of stories of these 568 buildings, 1-2 stories? Please add the photo taken date for Fig. 2. What are structures for the two houses in Fig. 2c and 2d? How did you confirm that the surveyed houses are the original damage condition or you evaluated the damage by asking all house owners? Was the almost damage from earthquake because of the retrofit/lesson from the 2010 event? It was possible to developed fragility curves for more than two damage levels. In Suppasri et al. (2012), they developed the curves for four damage levels using less than 200 buildings. You may change the word from "surviving" to other words such as remaining, not destroyed, etc. Buildings are not living things, thus the word "survive" sounds not so suitable in my opinion. Please add some photos of the buildings as example of your damage levels, destroyed and not destroyed. Washed away building should be also included in the "destroyed" damage level. No washed away buildings in this 2015 event? 3.2 Inundation depth: It is confusing if you used your measured inundation heights during the survey for your inundation simulation or just the recorded waveforms for the source verification? If you have all surveyed inundation heights for each building, you can then develop fragility functions using the actual maximum flow depth from the field survey and maximum flow velocity and hydrodynamic force by making use of the numerical simulation. Please make clear about this in this section. Please use the original reference for K and Kappa (Aida, 1978). 3.3 Fragility curves: Similar to section 3.2 that I would like to encourage the

authors to used their simulated maximum flow velocity and hydrodynamic force to also develop the fragility curves as successfully made by previous studies (Koshimura et al, 2009, Suppasri et al., (2011) and Gokon et al., (2014)). These curves using flow velocity and hydrodynamic force can be used to explain in your section 3.4 why your study area have lower damage probability (far and protected by railway embankment). Even if the same flow depth, but you may be able to explain that the flow velocity and hydrodynamic force is much lower in your area. 3.4 The title of the section is wrong? Should be Comparison of the developed fragility curves or something? I suggest to use bullets or subsection to compare your curves with 1) Dichato first, 2) then Okushiri and other part of Japan and 3) other parts of the world (Why didn't you also compare with Samoa and Thailand?) Can you spilt the data and develop another one or two curves for 1-story or 2-story only? Then you can show that the damage of the mix stories is lower than 1-story only or higher than 2-story only. 4.1 Source model: Did the event in 1849 a tsunami earthquake? Why such event is not necessary to include in this study? Please also write that you used earthquake magnitude (8.2-8.5) in the explanation not only in Fig. 9. 4.2 Proposed scenario: It will be very useful for readers if you could add one figure showing the comparison of the simulated flow depth of 2015 tsunami against the proposed scenario. May be the values shown in the map can be the difference between the proposed scenario and the 2015 tsunami (i.e. Fig. 10(center-S1) minus Fig. 7a). Then the readers can be easily seen that where we might expected greatly increase of the flow depth from the future event. Characteristics of the maximum flow velocity and hydrodynamic force can be also discussed and support your explanation at the end of this section 4.3 Tsunami mitigation measures are deeply discussed by Strusińska-Correia. (2017) and Suppasri et al. (2016)

Suggested references Agnieszka Strusińska-Correia. (2017) Tsunami mitigation in Japan after the 2011 Tōhoku Tsunami. International Journal of Disaster Risk Reduction 22, 397-411. Online publication date: 1-Jun-2017 Aida, I. (1978) Reliability of a tsunami source model derived from fault parameters, J. Phys. Earth, 26, 57–73. Charvet, I., Macabuag, J., Rossetto, T. (2017) Estimating tsunami-induced building

damage through fragility functions: Critical review and research needs, Front. Built Environ. 3:36. Fraser, S. A., Power, W. L., Wang, X., Wallace, L. M., Mueller, C. & Johnston, D. M. [2014] "Tsunami inundation in Napier, New Zealand, due to local earthquake sources," Nat. Hazards, 70(1), 415–445. Macabuag, J., Rossetto, T., Ioannou, I., Suppasri, A., Sugawara, D., Adriano, B., Imamura, F. and Koshimura, S. (2016) A proposed methodology for deriving tsunami fragility functions for buildings using optimum intensity measures, Natural Hazards, 84 (2), 1257-1285 Wiebe, D. M. & Cox, D. T. [2014] "Application of fragility curves to estimate building damage and economic loss at a community scale: A case study of Seaside, Oregon," Nat. Hazards 71(3), 2043–2061. Song, J., De Risi, R. and Goda, K. (2017) Influence of Flow Velocity on Tsunami Loss Estimation, Geosciences 2017, 7(4), 114. Suppasri, A., Latcharote, P., Bricker, J. D., Leelawat, N., Hayashi, A., Yamashita, K., Makinoshima, F., Roeber, V. and Imamura, F. (2016) Improvement of tsunami countermeasures based on lessons from the 2011 great east japan earthquake and tsunami -Situation after five years-, Coastal Engineering Journal, 58 (4), 1640011.

---

## Referee Comment (RC2) · Anonymous Referee #2 · 9 Dec 2017

This paper provides tsunami fragility curves for residential buildings in a town in Chile. The authors use a standard procedure, using model results to hindcast flow depth, then correlating with observations. Though no new methods are developed in this paper, the results are useful to engineers and planners in Chile and other tsunami-prone coasts. Therefore, it will be ready for publication after the authors address the following comments.

Title should be '... fragility curve development...'

Please have a native English speaker correct the manuscript, as there are many small mistakes. For example, on line 17, "low damage" should be "light damage", and similar

problems throughout the document.

Pae 2. Lines 17-18. You say that buildings in Sanriku experienced greater damage than those on the Sendai Plain. Is this just because the inundation was deeper in Sanriku? Or do you mean that for equal inundation depths, buildings in Sanriku experience more damage? Clarify this.

Page 3 line 15. Tsunami height of 7 m. Does this mean the height offshore? The height a the coastline? Runup? Please be specific.

Page 4 line 13. "The most affected structures were adobe. " You mean structures most affected by the earthquake? Clarify this.

Page 5 line 1. State DART buoy ID number.

Page 5, lines 10-15. State the source and publication numbers of the nautical charts used for bathymetry.

Page 5 line 16. What is the DTM resolution? What is its source (LIDAR or land surveys)? Does the DTM show pre-earthquake or post-earthquake (subsided) topography? How much tectonic subsidence occurred here?

Page 5 line 24. State the source (reference) of the JSCE guidelines.

Page 6 line 6. n=0.06 seems small for medium density urban areas. Looking at Table 1 in Bricker et al (which you already reference), the recommended values for medium density urban areas range from 0.09 (based on Koshimura et al) to 0.12 (based on Bunya et al). Therefore, you need to justify the value 0.06 you choose for medium density urban areas.

Page 6 line 31. What are the units of the mean and std dev values shown?

Page 7 lines 15-16. In addition to different building materials, could the difference in fragility curves have been due to the tsunami behavior? Bore vs. no bore? Flow speed? Impact of debris from the harbor? This section needs a little more discussion.

Figure 8. "probability of occurrence" is not a good label for the y-axis, as it appears to show CDF's of inunation depth, which is not the case. The axis label should be "probability of damage" or similar.

In some places in the text, you need to make superscripts for units (i.e., km2).

Section 4.2. What topography is used? Subsided or non-subsided? How much subsidence is expected for a quake of this size?

Figure 2. You should indicate what level of damage each photo corresponds to.

Fig 3. I don't understand what the red circles and yellow triangles in the figure on the right mean. You need a legend or a description in the caption.

Fg 7c caption. Plot (of what)? Needs better description.

Fig 10. Left center right are columns, not rows.

Fig 11. I don't understand what the blue shading colorscale means. You need to state this, or show a colorbar.

---

## Author Comment (AC1) · 30 Jan 2018

- Title: Please consider adding the part "application of the developed fragility curves" in your title R: the title will be modified according to suggestions of both reviewers, for example: "Development and application of tsunami fragility curve of the 2015 tsunami in Coquimbo, Chile".

- Abstract: Please also specify the earthquake magnitude of both the 2015 and future events as well as add a short explanation about the comparison of the developed curves against Dichato. R: the abstract will be modified according to suggestions, such as we add earthquake magnitudes and short explanation about the comparison with

fragility curve of Dichato.

1.- Introduction: -Please make sure that you use the same name, 2015 Illapel earthquake (same in Fig.1) throughout the paper. R: The name of the 2015 Illapel earthquake will be modified and check throughout the text.

- In the last sentence of this section, your objective is only about two sentences. I would suggest to write them clearly using bullets and separate them from the literature review. R: The introduction will be modified as suggested, such that we clearly write the objective in a separate paragraph from literature review.

- There are some recently published works about fragility curves and their application that should be mentioned: Song et al. (2017), Charvet et al. (2017), Macabuag et al. (2016), Suppasri et al. (2016), Fraser et al. (2014) and Wiebe and Cox (2014) R: recent papers on fragility curves and damages estimation suggested by the reviewer will be checked and included in both introduction and discussion.

2.- Study Area - Please also clearly state the topographical characteristic of the study area, plain area similar to Sendai? R: More topographical description will be added about the study area

3.- Developing fragility curve. - What kind of the field data used as I can only see the recorded waveforms? Did you also measure inundation heights/depths from the field survey? If so why didn't you used them for your model verification in addition to the recorded waveforms? R: We used both tsunami waveforms and inundation hight measurements. As explained in the paper, the tsunami waveforms from a DART buoy and two tidegauges were use to select the best tsunami source model. Then, the tsunami inundation heights measurements were used to select an appropriate roughness coefficient in order to fit both inundation heights and inundation area. We will explain better this methodology in the text in order to be sure the readers do not misunderstand it.

- 3.1 Data: What is the average numbers of stories of these 568 buildings, 1-2 stories?

R: The houses were 1-story buildings on average. Few of them were of 2 stories and the typical configuration is the first floor of masonry and the second floor of wood. Since the flow depth was less than 3 m, and it did not exceed in height 1-story buildings, we thought to analyze all structures as the same type of buildings. It will be easier the comparison with Dichato.

- Please add the photo taken date for Fig. 2. R: We will add the photo date in the Figure caption. All photos were taken on September 22nd of 2015, only 6 days after the event.

- What are structures for the two houses in Fig. 2c and 2d? R: we added a description of the structures in Figure 2. Both structures in 2c and d were originally made of confined masonry. The structure in figure 2c was being repaired at the moment of the field survey with masonry made of concrete blocks.

- How did you confirm that the surveyed houses are the original damage condition or you evaluated the damage by asking all house owners? Was the almost damage from earthquake because of the retrofit/lesson from the 2010 event? R: The field survey took place only 5 to 7 days after the event and just few of the structures were already under reconstruction, therefore, the damage observed was assumed to be the one caused by the tsunami. According to official reports, the damage to structures due to the earthquake was very limited in Coquimbo area, and most of the damage observed was non existing or light damage only (few cracks). To confirm this, the authors had the opportunity to compare damage on inundated and non inundated houses in Coquimbo, in order to be sure that the structural damage to inundated houses was due to the tsunami. This is another reason why we used two-level damage scale. Doing this, we also wanted to avoid including light damage (due to the earthquake) as tsunami damage. We will add some paragraphs to explain this procedure in the paper

- It was possible to developed fragility curves for more than two damage levels. In Suppasri et al. (2012), they developed the curves for four damage levels using less

than 200 buildings. R: Yes, it would have been possible. Unfortunately, when we made the field survey, the main goal was to develop the fragility curve to assess the damage in case of future event as well as to be compared with the existing fragility curve of another town in Chile, Dichato (Mas et al., 2012), which has only two damage levels. Therefore, we classified the structures according to two damage levels only. In addition, as explained above, since the earthquake generated no or light damage to structure, the use of two damage levels was to avoid the classification of light damage due to earthquake as tsunami damage.

- You may change the word from "surviving" to other words such as remaining, not destroyed, etc. Buildings are not living things, thus the word "survive" sounds not so suitable in my opinion. R: The word "surviving" was changed by "not destroyed"

- Please add some photos of the buildings as example of your damage levels, destroyed and not destroyed. Washed away building should be also included in the "destroyed" damage level. No washed away buildings in this 2015 event? R. We will mention the damage level of the structures in figure 2. In addition, there were some washed away building, and we will include more pictures of those cases.

- 3.2 Inundation depth: It is confusing if you used your measured inundation heights during the survey for your inundation simulation or just the recorded waveforms for the source verification? If you have all surveyed inundation heights for each building, you can then develop fragility functions using the actual maximum flow depth from the field survey and maximum flow velocity and hydrodynamic force by making use of the numerical simulation. R: Unfortunately, the surveyed inundation heights were not for each building. The measurements were taken in several places in order to interpolate the flow depth as it was made in the paper of Mas et al., (2012) on Dichato. However, due to the absence of the tsunami traces on the wetland and at locations of washed away houses, it was not possible to obtain a representative interpolated inundation area. Subsequently, we decided to run tsunami numerical simulations and validate the results by means of the K and kappa coefficients using our tsunami height

measurements. We will explain better this methodology in order to avoid confusion to the readers.

- Please make clear about this in this section. Please use the original reference for K and Kappa (Aida, 1978). R: we will use the proper reference as Aida 1978.

- 3.3 Fragility curves: Similar to section 3.2 that I would like to encourage the authors to used their simulated maximum flow velocity and hydrodynamic force to also develop the fragility curves as successfully made by previous studies (Koshimura et al, 2009, Suppasri et al., (2011) and Gokon et al., (2014)). These curves using flow velocity and hydrodynamic force can be used to explain in your section 3.4 why your study area have lower damage probability (far and protected by railway embankment). Even if the same flow depth, but you may be able to explain that the flow velocity and hydrodynamic force is much lower in your area. R: As recommended by the reviewer, we added new fragility curves using flow velocity and hydrodynamic force. In addition, we also added a map of maximum flow velocity and hydrodynamic force in Coquimbo area in order to explain better the tsunami damage.

- 3.4 The title of the section is wrong? Should be Comparison of the developed fragility curves or something? R: Yes, the title was wrong and it was changed to the correct one: "Comparison with existing fragility curves"

- I suggest to use bullets or subsection to compare your curves with 1) Dichato first, 2) then Okushiri and other part of Japan and 3) other parts of the world (Why didn't you also compare with Samoa and Thailand?) R: we will improve the subsection according to the suggestion. We did not use the curves of Samoa and Thailand due to the fact that the building material was different (RC). We only used curves from similar construction material. We will add a sentence to explain this in the text.

- Can you spilt the data and develop another one or two curves for 1-story or 2-story only? Then you can show that the damage of the mix stories is lower than 1-story only or higher than 2-story only. R: We agreed this analysis could be useful, unfortunately,

we did not classify the structures according to the stories since they were very similar in size and material. In addition, we wanted to compare the results with the fragility curve of Dichato, which also considers one type of structure. Moreover, the main task of the paper was to develop a fragility function to assess tsunami damage in case of possible event in the future. The comparison of several curves according to number of stories could be out of the scope of the paper. However, we could perfectly add a comment on this based on existing literature.

- 4.1 Source model: Did the event in 1849 a tsunami earthquake? Why such event is not necessary to include in this study? R: R: we added details on the 1849 event in the current draft, and this event is very important when segments of possible events were defined. Unfortunately there is no enough information about this event, thus a source model is defined.

- Please also write that you used earthquake magnitude (8.2-8.5) in the explanation not only in Fig. 9. R: We will add the magnitudes of the possible events in the text and not only in the figure 9.

- 4.2 Proposed scenario: It will be very useful for readers if you could add one figure showing the comparison of the simulated flow depth of 2015 tsunami against the proposed scenario. May be the values shown in the map can be the difference between the proposed scenario and the 2015 tsunami (i.e. Fig. 10(center-S1) minus Fig. 7a). Then the readers can be easily seen that where we might expected greatly increase of the flow depth from the future event. R: good idea. We will add another box in Figure 11 to show the difference of 2015 event and the S1 event.

- Characteristics of the maximum flow velocity and hydrodynamic force can be also discussed and support your explanation at the end of this section R: Yes, since we will include new fragility curves using velocities and hydrodynamic forces as well as maximum velocity field maps, we will add more discussion based on these results and existing literature.

4.3 Tsunami mitigation measures are deeply discussed by Strusinska-Correia. (2017) and Suppasri et al. (2016). R: we will add some sentence on the possible mitigation measures based on the literature suggested by the reviewer.
* * *

---

## Author Comment (AC2) · 30 Jan 2018

- Title should be '... fragility curve development...' R: the title will be modified according to suggestions of both reviewers, for example: "Development and application of tsunami fragility curve of the 2015 tsunami in Coquimbo, Chile".

- Please have a native English speaker correct the manuscript, as there are many small mistakes. For example, on line 17, "low damage" should be "light damage", and similar problems throughout the document. R: the new draft will be corrected by a native English speaker in order to avoid small mistakes.

[Figure]

- Pae 2. Lines 17-18. You say that buildings in Sanriku experienced greater damage than those on the Sendai Plain. Is this just because the inundation was deeper in Sanriku? Or do you mean that for equal inundation depths, buildings in Sanriku experience more damage? Clarify this. R: Since we will add new fragility curves using velocity and hydrodynamic force as well as maximum velocity field, it will help clarifying why areas with equal inundation depth could have more damage. It is expected that plain areas have lower velocities than ria coasts.

- Page 3 line 15. Tsunami height of 7 m. Does this mean the height offshore? The height a the coastline? Runup? Please be specific. R: according to the references, it was 7 m inundation height. We will be more specific in the manuscript.

- Page 4 line 13. "The most affected structures were adobe. " You mean structures most affected by the earthquake? Clarify this. R: yes, we will correct the sentence by "the most affected structures due to the earthquake were made of adobe..."

- Page 5 line 1. State DART buoy ID number. R: The DART buoy is the 32402. We will add the ID number.

- Page 5, lines 10-15. State the source and publication numbers of the nautical charts used for bathymetry. R: ok, we will add the detail of the publication numbers if the nautical charts.

- Page 5 line 16. What is the DTM resolution? What is its source (LIDAR or land surveys)? Does the DTM show pre-earthquake or post-earthquake (subsided) topography? How much tectonic subsidence occurred here? R: We will add details on the DTM from LIDAR topography provided by the local authority. We will also add some comment of the subsidence due to the 2015 earthquake, which was in the order of 5 cm.

- Page 5 line 24. State the source (reference) of the JSCE guidelines. R: we will use the original reference (Aide 1978) as suggested by Referee 1.

- Page 6 line 6. n=0.06 seems small for medium density urban areas. Looking at Table 1 in Bricker et al (which you already reference), the recommended values for medium density urban areas range from 0.09 (based on Koshimura et al) to 0.12 (based on Bunya et al). Therefore, you need to justify the value 0.06 you choose for medium density urban areas. R: We use n=0.06 based on Kotani et al (1998) given in Table 1. We did not try larger values, since the larger the manning coefficient, the smaller the inundation. In addition, the best results was for n=0.025.

- Page 6 line 31. What are the units of the mean and std dev values shown? R. The units are metres. It will be added to the paper.

- Page 7 lines 15-16. In addition to different building materials, could the difference in fragility curves have been due to the tsunami behavior? Bore vs. no bore? Flow speed? Impact of debris from the harbor? This section needs a little more discussion. R: Yes, more discussion will be added since fragility curve using velocity and maximum velocity field will be included in the new version of the paper. It will help to understand the tsunami behavior and impact in Coquimbo.

-Figure 8. "probability of occurrence" is not a good label for the y-axis, as it appears to show CDF's of inunation depth, which is not the case. The axis label should be"probability of damage" or similar. R: Yes, the y-label is not appropriate. It will be changed to Damage probability.

- In some places in the text, you need to make superscripts for units (i.e., km2). R: ok, it will be checked throughout the text.

- Section 4.2. What topography is used? Subsided or non-subsided? How much subsidence is expected for a quake of this size? R: According to Figure 10, the subsidence is <1m. It is indeed included in the numerical simulation. We will add a sentence with the subsidence of each tsunami scenario, which is already included in the inundation simulation.

- Figure 2. You should indicate what level of damage each photo corresponds to. R: yes, it will be added. In addition, as suggested by Referee 1, we will added photos of washed away, too.

- Fig 3. I don't understand what the red circles and yellow triangles in the figure on the right mean. You need a legend or a description in the caption. R: We will add a description in the caption. Triangles and circles are runup and inundation heights measurements, respectively.

- Fg 7c caption. Plot (of what)? Needs better description. R: the plot of least-squares fit will be added as a description to the figure 7c.

- Fig 10. Left center right are columns, not rows. R: it will be corrected in the figure caption

Fig 11. I don't understand what the blue shading colorscale means. You need to state this, or show a colorbar R: the caption of the figure will be modified in order to explain better the inundation area due to the possible tsunami scenario as follows: Figure 11. a) Tsunami flow depth on structures. b) Damage probability of flooded structures. The light blue area represents the tsunami inundation area due to the possible future tsunami.

---

## Author Response (AR1)

**Response to reviewer**

The following document describes the response to reviewer. All comments and questions were separated by topics thus point-by-point reviews are included.
In addition, this document includes a marked-up version of the manuscript with all changes made according to reviewers suggestions.

**Reviewer 1**
- Title: Please consider adding the part "application of the developed fragility curves" in your title
*R: the title will be modified according to suggestions of both reviewers, for example: "Development and application of tsunami fragility curve of the 2015 tsunami in Coquimbo, Chile".*

- Abstract: Please also specify the earthquake magnitude of both the 2015 and future events as well as add a short explanation about the comparison of the developed curves against Dichato.
*R: we added earthquake magnitudes and short description on the comparison with fragility curve of Dichato.*

1.- Introduction:
-Please make sure that you use the same name, 2015 Illapel earthquake (same in Fig.1) throughout the paper.
*R: The name of the 2015 Illapel earthquake will be modified and check throughout the text.*

- In the last sentence of this section, your objective is only about two sentences. I would suggest to write them clearly using bullets and separate them from the literature review.
*R: The introduction will be modified, such that we clearly write the objective in a separate paragraph from literature review. In addition, we added description of the sections of the paper.*

- There are some recently published works about fragility curves and their application that should be mentioned: Song et al. (2017), Charvet et al. (2017), Macabuag et al. (2016), Suppasri et al. (2016), Fraser et al. (2014) and Wiebe and Cox (2014)
*R: recent papers on fragility curves and damages estimation suggested by the reviewer were checked and included in both introduction and discussion.*

2.- Study Area
- Please also clearly state the topographical characteristic of the study area, plain area similar to Sendai?
*R: the paper includes a topographical description of the study area*

3.- Developing fragility curve.
- What kind of the field data used as I can only see the recorded waveforms? Did you also measure inundation heights/depths from the field survey? If so why didn't you used them for your model verification in addition to the recorded waveforms?
*R: We used both tsunami waveforms and inundation height measurements. As explained in the paper, the tsunami waveforms from a DART buoy and two tidegauges were used to select the best tsunami source model. Then, the tsunami inundation heights measurements were used to select an appropriate roughness coefficient in order to fit both inundation heights and inundation area. We explained better this methodology in the text in order to be sure the readers do not misunderstand it.*

- 3.1 Data: What is the average numbers of stories of these 568 buildings, 1-2 stories?
*R: The houses were 1-story buildings on average. some of them were of 2 stories and the typical configuration is the first floor of masonry and the second floor of wood. Since the flow depth was less than 3 m, and it did not exceed in height 1-story buildings, we thought to analyze all structures as the same type of buildings. It will be easier the comparison with Dichato.*

- Please add the photo taken date for Fig. 2.
*R: We will add the photo date in the Figure caption. All photos were taken on September 22nd of 2015, only 6 days after the event.*

- What are structures for the two houses in Fig. 2c and 2d?
*R: we added a description of the structures in Figure 2. Both structures in 2c and d were originally made of confined masonry. The structure in figure 2c was being repaired at the moment of the field survey with masonry made of concrete blocks.*

- How did you confirm that the surveyed houses are the original damage condition or you evaluated the damage by asking all house owners? Was the almost damage from earthquake because of the retrofit/lesson from the 2010 event?
*R: The field survey took place only 5 to 7 days after the event and just few of the structures were already under reconstruction, therefore, the damage observed was assumed to be the one caused by the tsunami. According to official reports, the damage to structures due to the earthquake was very limited in Coquimbo area. To confirm this, the authors had the opportunity to compare damage on inundated and non inundated houses in Coquimbo, in order to be sure that the structural damage to inundated houses was due to the tsunami. This is another reason why we used two-level damage scale. Doing this, we also wanted to avoid including light damage (possibly due to the earthquake) as tsunami damage. We will add some paragraphs to explain this procedure in the paper*

- It was possible to developed fragility curves for more than two damage levels. In Suppasri et al. (2012), they developed the curves for four damage levels using less than 200 buildings.
*R: Yes, it would have been possible. Unfortunately, when we made the field survey, the main goal was to develop the fragility curve to assess the damage in case of future event as well as to be compared with the existing fragility curve of another town in Chile, Dichato (Mas et al., 2012), which has only two damage levels. Therefore, we classified the structures according to two damage levels only. In addition, as explained above, since the earthquake generated no or light damage to structure, the use of two damage levels was to avoid the classification of light damage due to earthquake as tsunami damage.*

- You may change the word from "surviving" to other words such as remaining, not destroyed, etc. Buildings are not living things, thus the word "survive" sounds not so suitable in my opinion.
*R: The word "surviving" was changed by "not destroyed"*

- Please add some photos of the buildings as example of your damage levels, destroyed and not destroyed. Washed away building should be also included in the "destroyed" damage level. No washed away buildings in this 2015 event?
*R. We mentioned damage levels of the structures in figure 2. In addition, we included more pictures of complete damage and wased away.*

- 3.2 Inundation depth: It is confusing if you used your measured inundation heights during the survey for your inundation simulation or just the recorded waveforms for the source verification? If you have

all surveyed inundation heights for each building, you can then develop fragility functions using the actual maximum flow depth from the field survey and maximum flow velocity and hydrodynamic force by making use of the numerical simulation.

*R: Unfortunately, the surveyed inundation heights were not for each building. The measurements were taken in several places in order to interpolate the flow depth as it was made in the paper of Mas et al., (2012) on Dichato. However, due to the absence of the tsunami traces on the wetland and at locations of washed away houses, it was not possible to obtain a representative interpolated inundation area. Subsequently, we decided to run tsunami numerical simulations and validate the results by means of the K and kappa coefficients using our tsunami height measurements. We added a sentence in the manuscript to explain this methodology in order to avoid confusion to the readers.*

- Please make clear about this in this section. Please use the original reference for K and Kappa (Aida, 1978).

*R: we will use the proper reference as Aida 1978.*

- 3.3 Fragility curves: Similar to section 3.2 that I would like to encourage the authors to used their simulated maximum flow velocity and hydrodynamic force to also develop the fragility curves as successfully made by previous studies (Koshimura et al, 2009, Suppasri et al., (2011) and Gokon et al., (2014)). These curves using flow velocity and hydrodynamic force can be used to explain in your section 3.4 why your study area have lower damage probability (far and protected by railway embankment). Even if the same flow depth, but you may be able to explain that the flow velocity and hydrodynamic force is much lower in your area.

*R: As recommended by the reviewer, we added new fragility curves using flow velocity and hydrodynamic force. In addition, we also added a map of maximum flow velocity and hydrodynamic force in Coquimbo area in order to explain better the tsunami damage.*

- 3.4 The title of the section is wrong? Should be Comparison of the developed fragility curves or something?

*R: Yes, the title was wrong and it was changed to the correct one: "Comparison with existing fragility curves"*

- I suggest to use bullets or subsection to compare your curves with 1) Dichato first, 2) then Okushiri and other part of Japan and 3) other parts of the world (Why didn't you also compare with Samoa and Thailand?)

*R: Since we compare different types of curves, and a group of them was made considering only one damage level (Okushiri, Dichato, Banda Aceh, Thailnad and Samoa) we think it would be more confusing to compare by country. The focus is the structural type and topography.. Moreover, We added fragility curves of Samoa and Thailand in the new version of the draft.*

- Can you spilt the data and develop another one or two curves for 1-story or 2-story only? Then you can show that the damage of the mix stories is lower than 1-story only or higher than 2-story only.

*R: We agreed this analysis could be useful, unfortunately, we did not classify the structures according to the stories since they were very similar in size and material. In addition, we wanted to compare the results with the fragility curve of Dichato, which also considers one type of structure independently of the number of stories. Moreover, the main task of the paper was to develop a fragility function to assess*

*tsunami damage in case of possible event in the future. The comparison of several curves according to number of stories could be out of the scope of the paper.*

- 4.1 Source model: Did the event in 1849 a tsunami earthquake? Why such event is not necessary to include in this study?
*R: we added details on the 1849 event in the current draft, and this event is very important when segments of possible events were defined. Unfortunately there is no enough information about this event, thus a source model is defined.*

- Please also write that you used earthquake magnitude (8.2-8.5) in the explanation not only in Fig. 9.
*R: We will add the magnitudes of the possible events in the text and not only in the figure.*

- 4.2 Proposed scenario: It will be very useful for readers if you could add one figure showing the comparison of the simulated flow depth of 2015 tsunami against the proposed scenario. May be the values shown in the map can be the difference between the proposed scenario and the 2015 tsunami (i.e. Fig. 10(center-S1) minus Fig. 7a). Then the readers can be easily seen that where we might expected greatly increase of the flow depth from the future event.
*R: good idea. We will add a new figure to show results and the difference of 2015 event and the S1 event.*

- Characteristics of the maximum flow velocity and hydrodynamic force can be also discussed and support your explanation at the end of this section
*R: Yes, since we will include new fragility curves using velocities and hydrodynamic forces as well as maximum velocity field maps, we will add more discussion based on these results and existing literature.*

4.3 Tsunami mitigation measures are deeply discussed by Strusinska-Correia. (2017) and Suppasri et al. (2016).
*R: we added some sentence on the possible mitigation measures based on the literature suggested by the reviewer. We also added other references with local analysis after 2010 event.*

Suggested references:

Agnieszka Strusinska-Correia. (2017) Tsunami mitigation in Japan after the 2011 Tohoku Tsunami. International Journal of Disaster Risk Reduction 22, 397-411. Online publication date: 1-Jun-2017

Aida, I. (1978) Reliability of a tsunami source model derived from fault parameters, J. Phys. Earth, 26, 57–73.

Charvet, I., Macabuag, J., Rossetto, T. (2017) Estimating tsunami-induced building damage through fragility functions: Critical review and research needs, Front. Built
Environ. 3:36.

Fraser, S. A., Power, W. L., Wang, X., Wallace, L. M., Mueller, C. & Johnston, D. M. [2014] "Tsunami inundation in Napier, New Zealand, due to local earthquake sources," Nat. Hazards, 70(1), 415–445.

Macabuag, J., Rossetto, T., Ioannou, I., Suppasri, A., Sugawara, D., Adriano, B., Imamura, F. and Koshimura, S. (2016). A proposed methodology for deriving tsunami fragility functions for buildings using optimum intensity measures, Natural Hazards, 84 (2), 1257-1285

Wiebe, D. M. & Cox, D. T. [2014] "Application of fragility curves to estimate building damage and economic loss at a community scale: A case study of Seaside, Oregon," Nat. Hazards 71(3), 2043–2061.

Song, J., De Risi, R. and Goda, K. (2017) Influence of Flow Velocity on Tsunami Loss Estimation, Geosciences 2017, 7(4), 114.

Suppasri, A., Latcharote, P., Bricker, J. D., Leelawat, N., Hayashi, A., Yamashita, K., Makinoshima, F., Roeber, V. and Imamura, F. (2016) Improvement of tsunami countermeasures based on lessons from the 2011 great east japan earthquake and tsunami -Situation after five years-, Coastal Engineering Journal, 58 (4), 164

**Reviewer 2**

- Title should be '... fragility curve development...'
*R: the title was modified according to suggestions of both reviewers, for example: "Development and application of tsunami fragility curve of the 2015 tsunami in Coquimbo, Chile".*

- Please have a native English speaker correct the manuscript, as there are many small mistakes. For example, on line 17, "low damage" should be "light damage", and similar problems throughout the document.
*R: the new draft was corrected by a native English speaker in order to avoid small mistakes.*

- Pae 2. Lines 17-18. You say that buildings in Sanriku experienced greater damage than those on the Sendai Plain. Is this just because the inundation was deeper in Sanriku? Or do you mean that for equal inundation depths, buildings in Sanriku experience more damage? Clarify this.
*R: Since we added new fragility curves using velocity and hydrodynamic force as well as maximum velocity field, it helped clarifying why areas with equal inundation depth could have more damage. It is expected that plain areas have smaller velocities than ria coasts.*

- Page 3 line 15. Tsunami height of 7 m. Does this mean the height offshore? The height a the coastline? Runup? Please be specific.
*R: according to the references, it was 7 m inundation height. We were more specific in the manuscript.*

- Page 4 line 13. "The most affected structures were adobe. " You mean structures most affected by the earthquake? Clarify this.
*R: yes, we corrected the sentence by "the most affected structures by the earthquake were made of adobe..."*

- Page 5 line 1. State DART buoy ID number.
*R: The DART buoy is the 32402. We will add the ID number.*

- Page 5, lines 10-15. State the source and publication numbers of the nautical charts used for bathymetry.
*R: ok, we added the detail of the publication numbers if the nautical charts.*

- Page 5 line 16. What is the DTM resolution? What is its source (LIDAR or land surveys)? Does the DTM show pre-earthquake or post-earthquake (subsided) topography? How much tectonic subsidence occurred here?
*R: We add details on the DTM topography provided by the local authority. We will also add some comment of the subsidence due to the 2015 earthquake, which was in the order of 7-10 cm.*

- Page 5 line 24. State the source (reference) of the JSCE guidelines.
*R: we used the original reference (Aide 1978) as suggested by Referee 1.*

- Page 6 line 6. n=0.06 seems small for medium density urban areas. Looking at Table 1 in Bricker et al (which you already reference), the recommended values for medium density urban areas range from 0.09 (based on Koshimura et al) to 0.12 (based on

Bunya et al). Therefore, you need to justify the value 0.06 you choose for medium density urban areas.
*R: We used n=0.06 based on Kotani et al (1998) given in Table 1. We did not try larger values, since the larger the manning coefficient, the smaller the inundation. In addition, the best results was for n=0.025, therefore, there is no effect on the final result.*

- Page 6 line 31. What are the units of the mean and std dev values shown?
*R. The units are metres. It will be added to the paper.*

- Page 7 lines 15-16. In addition to different building materials, could the difference in fragility curves have been due to the tsunami behavior? Bore vs. no bore? Flow speed? Impact of debris from the harbor? This section needs a little more discussion.
*R: Since we added other fragility curves and compare the results with other places with plain-type or ria-type coasts, it is possible to explain different tsunami effect. This analysis was included in the new manuscript..*

-Figure 8. "probability of occurrence" is not a good label for the y-axis, as it appears to show CDF's of inunation depth, which is not the case. The axis label should be"probability of damage" or similar.
*R: Yes, the y-label is not appropriate. It will be changed to Damage probability.*

- In some places in the text, you need to make superscripts for units (i.e., km2).
*R: ok, it was checked throughout the text.*

- Section 4.2. What topography is used? Subsided or non-subsided? How much subsidence is expected for a quake of this size?
*R: We added a sentence with the subsidence of S1 scenario, which was already included in the inundation simulation.*

- Figure 2. You should indicate what level of damage each photo corresponds to.
*R: yes, it was added. In addition, as suggested by Referee 1, we will added photos of washed away, too.*

- Fig 3. I don't understand what the red circles and yellow triangles in the figure on the right mean. You need a legend or a description in the caption.
*R: We added a description in the caption. Triangles and circles are runup and inundation heights measurements, respectively.*

- Fg 7c caption. Plot (of what)? Needs better description.
*R: the plot of least-squares fit will be added as a description to the figure 7c. it is corrected in new figure 8b*

- Fig 10. Left center right are columns, not rows.
*R: it will be corrected in the figure caption*

Fig 11. I don't understand what the blue shading colorscale means. You need to state this, or show a colorbar
*R: the caption of the figure will be modified in order to explain better the inundation area due to the possible tsunami scenario as follows: Figure 11. a) Tsunami flow depth on structures. b) Damage probability of flooded structures. The light blue area represents the tsunami inundation area due to the possible future tsunami.*

[revised manuscript text omitted]

---

## Author Response (AR2)

**Response to reviewers**

The following document describes the response to reviewers. In addition, this document includes a marked-up version of the manuscript with all changes made according to reviewers suggestions.

**Reviewer #2**
The author has responded sufficiently to my comments. The only remaining item is the English of the paper, which still needs to be improved for publication in NHESS.

**R:** The English of the manuscript was revised and improved by a Native English speaker.

[revised manuscript text omitted]